# Continuous Heterogeneous Fenton for Swine Wastewater Treatment: Converting an Industry Waste into a Wastewater Treatment Material

**João Lincho** [ID], **João Gomes** [ID], **Rui C. Martins** [ID] and **Eva Domingues** *[ID]

University of Coimbra, CERES, Department of Chemical Engineering, Faculty of Sciences and Technology, Rua Sílvio Lima, Polo II, 3030-790 Coimbra, Portugal; jlincho@eq.uc.pt (J.L.); jgomes@eq.uc.pt (J.G.); martins@eq.uc.pt (R.C.M.)
* Correspondence: evadomingues@eq.uc.pt

**Abstract:** Swine wastewater (SW) was treated using industrial wastes as raw materials in a pre-treatment process (coagulation or adsorption), followed by a continuous heterogeneous Fenton reaction. Before the treatment conducted as a continuous operation, two different batch optimization strategies were evaluated, in which the effects of $H_2O_2$ concentration and pH were studied. The results show that using excessive $H_2O_2$ results in the same behavior, regardless of whether the pH is 3 or 7.5, while at low $H_2O_2$ concentrations, the acidic pH improves the chemical oxygen demand (COD) removal due to a higher solubility of iron. The partial addition of $H_2O_2$ after 60 min of the reaction proved to be unbeneficial. Considering other perspectives, a continuous Fenton process using iron filings (IF) as the iron source ($[H_2O_2]$ = 50 mg/L) was applied after the SW pre-treatment, consisting of adsorption with red mud (RM) or coagulation with poly-diallyldimethylammonium chloride (PDADMAC). The RM adsorption presented higher COD removal and lower toxicity than the PDADMAC coagulation, revealing to be a suitable material for this purpose, but for both pre-treatments, the application of a subsequent continuous Fenton process revealed to be essential to achieve the COD discharge limits imposed by the Portuguese law. In addition, high amounts of dissolved iron were present in the samples (55–58 mg/L) after the Fenton process. However, after the overall treatment, the samples showed no harmful characteristics for *Lepidium sativum*, being classified as "non-toxic", contrary to the initial wastewater.

**Keywords:** continuous Fenton process; heterogeneous Fenton; iron filings; red mud; swine wastewater



## 1. Introduction

The world's population growth is contributing to a high demand in the food market [1–3]. Since pork meat is the second most consumed meat, the swine industry is growing quickly [4,5]. About 1 million pigs, corresponding to 102 million tons of meat, are produced per year. China leads the production with approximately 450 million pigs, followed by the European Union with 140 million, and the United States of America with 74 million heads in April 2022 [6–8].

The intensive meat production industry has critical consequences for public health and the environment due to the residues that are generated, namely animal wastes, contaminated wastewater, and the release of different kinds of gases [4]. Moreover, this livestock production results in high water consumption (an estimated of 6 $m^3$/kg of pork meat in Brazil), with the swine industry being responsible for about 43% of the generated livestock wastewater [4,8,9]. The swine wastewater (SW) is complex, being characterized by high organic loads due to the existence of fatty acids, nitrogen, phosphorous, heavy metals, antibiotics, suspended solids, and fecal coliforms [1,4,6,9]. Therefore, this kind of wastewater can be dangerous due to the existence high loads of organic matter (OM), ammonium and ammonia, antibiotics, and pathogenic organisms [4,5,10–12].

Natural organic matter (NOM) occurs due to the residues from plants, animals, and humic substances, while synthetic organic matter (SOM) is usually complex and dangerous, and arising from the use of chemicals in industries or agriculture [13]. The high organic loads of swine wastewater are a threat to the environment if not properly treated. Phosphorous and nitrogen are macronutrients that can lead to the eutrophication phenomenon [14,15]. Ammonium is usually present at high concentrations in swine wastewater [16], and although it does not present toxic effects, it can cause odors and microbial development [17]. Ammonia can cause the acidification of water, eutrophication, or toxicity and other harmful effects on aquatic organisms [18]. Moreover, SW can present high levels of virus and protozoan agents, which can be sources of diseases [4], therefore, the disinfection of this wastewater should also be considered.

Antibiotics are used to treat diseases in animals and are usually poorly metabolized, with 30–90% being excreted as parent compounds or metabolites [19,20]. Urine and feces are usually treated and used as fertilizers in agriculture, which can be a source of soil contamination and water pollution due to the presence of pharmaceuticals [21]. In 2013, about 52% of the total consumption of antibiotics in China was attributed to animals, and about 50% entered the environment [21,22]. Several antibiotics have been found in rivers worldwide [11], with one contributing factor being the incapacity of industrial and urban wastewater treatment plants (WWTPs) to remove them before the (treated) wastewater discharge. There is also the risk of increasing antibiotic resistance due to the widespread and extensive use of antibiotics, which increases concerns for public health and the natural environment [23,24]. Therefore, it is important to ensure the proper swine wastewater treatment.

Water scarcity is also a problem that already affects the world, in which about 1.2 billion people already suffer from lack of water, with this number being expected to increase to about 1.8 billion people in 2025 [25,26]. In addition, about 2.7 billion people face water scarcity at least one month per year, and about 2.4 billion people are exposed to diseases (such as cholera or typhoid fever) due to inadequate sanitation [26]. Therefore, the correct water/wastewater management (with the correct water reuse) must be addressed, since it can mitigate the water scarcity and contamination problems [9,25]. Several countries have implemented policies to encourage proper SW management and treatment [8]. The SW treatment is usually done by biological systems (such as aerobic lagoons, anaerobic lagoons, anaerobic packed-bed reactors, anaerobic filters, anaerobic bioreactors), but these conventional methods may not be capable of reaching the legal limits for discharge [9]. In fact, some of these methods are extremely popular, presenting economic feasibility and energy recovery capability, but they also require high retention times and are very sensitive to the process conditions [4].

Advanced oxidation processes (AOPs) are complementary technologies for industrial wastewater treatment since they offer high reliability in the contaminant's degradation. AOPs can be used either after or before biological processes, complementing the treatment quality. In this perspective, AOPs can be applied after the biological treatment to remove recalcitrant contaminants that the biological route was not capable of removing, while when used before, they can reduce the wastewater toxicity and improve the efficiency of the forthcoming biological treatment [27]. In this context, the Fenton reaction can be a suitable solution due to its simplicity and high efficiency in the removal of harmful pollutants from water and wastewater.

The Fenton reaction uses $Fe^{2+}$ to initiate and catalyze $H_2O_2$ decomposition, forming hydroxyl radicals ($\bullet OH$) and $Fe^{3+}$, and, due to the action of $H_2O_2$, at the same time the $Fe^{3+}$ is reduced to $Fe^{2+}$, forming the hydroperoxyl radical ($\bullet HO_2$), allowing the chain reaction to continue [28,29]. Both $Fe^{2+}$ and the $H_2O_2$ can also be $\bullet OH$ scavengers. In excess, the $Fe^{2+}$ can react with the $\bullet OH$ radicals and form $Fe^{3+}$ and $OH^-$ [29,30], while the $H_2O_2$ can produce $\bullet HO_2$ or self-decompose in water and oxygen. When the amount of $H_2O_2$ is low, it can cause low radical generation, decreasing the Fenton efficiency [31,32]. The Fenton key parameters are the concentration of Fe and $H_2O_2$, pH, temperature, and the

concentration of organic and inorganic species, and to obtain the maximum performance, it is also necessary to understand the relation between the $[Fe^{2+}]/[H_2O_2]$ ratio and •OH production and consumption [33].

Typically, the Fenton reaction occurs as a homogeneous system [28], and often at low pH, with the best reported pH in literature being 3–4 [33,34]. A neutral pH can also be used, but the efficiency is lower [35], and at a basic pH, the Fenton reaction is limited, since the iron precipitates in the $Fe(OH)_3$ form [36]. Additionally, at a high pH, the hydrogen peroxide can decompose in oxygen and water. Other metals, such as Cu, Ce, Mn, Cr Co, Ru, or Al can be used in processes known as Fenton-like processes [29], and although Fenton-like reactions can be used in neutral or basic pH, they can present disadvantages, such as high-costs, metal leaching, complex mechanisms, and low catalyst reuse [29]. Iron sludge production is a common drawback of homogeneous Fenton, but the use of heterogeneous materials could overcome this issue. Using zero valent iron as a heterogeneous iron source can be advantageous, since it is cheap and widely available (iron powders, filings, wires, nails, wool, or nanoparticles), simple and easy to handle [37]. The $Fe^0$ can be oxidized to $Fe^{2+}$ by different mechanisms (Equations (1)–(4)), generating •OH or other radicals in the classic Fenton reaction mechanism [37,38]. However, the use of $Fe^0$ has the unique advantage of being capable of regenerating the $Fe^{3+}$ into $Fe^{2+}$ at the iron surface (Equation (5)), being a cost-saving process when compared to the typical regeneration of $Fe^{3+}$ by $H_2O_2$ action [39,40].

$$Fe^0 \rightarrow Fe^{2+} + 2\,e^- \tag{1}$$

$$Fe^0 + H_2O \rightarrow Fe^{2+} + H_2 + 2\,OH^- \tag{2}$$

$$2\,Fe^0 + O_2 + 2\,H_2O \rightarrow 2\,Fe^{2+} + 4\,OH^- \tag{3}$$

$$Fe^0 + H_2O_2 \rightarrow Fe^{2+} + 2\,OH^- \tag{4}$$

$$Fe^0 + 2\,Fe^{3+} \rightarrow 3\,Fe^{2+} \tag{5}$$

Fenton can present several advantages when compared to other AOPs. It is characterized by low-cost, simplicity and low toxicity of the required materials [41]. When coupled with radiation (photo-Fenton) it can increase the efficiency and decrease the required amount of $Fe^{2+}$ and $H_2O_2$ [34,42–46]. The presence of radiation favors the reduction of $Fe^{3+}$ into $Fe^{2+}$ also forming •OH (Equation (6)) [37]. Using the sun as a radiation source can be an interesting alternative to promote photo-Fenton without increasing energy costs [36,46–48].

$$Fe^{3+} + H_2O + hv \rightarrow Fe^{2+} + H^+ + •OH \tag{6}$$

Regarding the other technologies, ozonation can be efficient in removing color, odor, taste, and pollutants from wastewater, but the action of ozone molecules is selective, being associated with low gas/liquid transference rates and poor COD and TOC removal. Moreover, it can present a short lifetime of ozone and a high energy demand as drawbacks [36,49]. Combining ozone with hydrogen peroxide (the peroxone process) can enhance the reaction kinetics, improving the •OH generation and reducing the amount of required ozone, but the reaction initiation step is slow [29,30]. Combining radiation with $H_2O_2$ can also be used for •OH generation (Equation (7)), but this reaction has a low quantum yield and requires UVC radiation [30,37].

$$H_2O_2 + hv \rightarrow 2\,•OH \tag{7}$$

Photocatalysis requires radiation and a photocatalyst, and it is usually associated with low reaction kinetics and high recombination rates [34,50,51]. Adding $H_2O_2$ to photocatalysis allows the generation of additional •OH radicals due to the better trapping of conduction band electrons or the reaction with superoxide radicals [37]. Unfortunately, the traditional materials used in photocatalysis are in powder form, which implies difficult operations for recovering the photocatalyst, posing an obstacle to industrial application [52]. Therefore, Fenton can be a suitable technology over other advanced oxidation processes, and by using industrial wastes as the iron source, it increases the economic feasibility of

Fenton while maintaining high efficiency, simplicity, and easy separation of the treated liquid from the iron source.

Using homogenous Fenton with $FeSO_4$, Lee et al. [53] removed 86% of COD from a SW with 5000–5700 $mgO_2/L$ using $[Fe^{2+}] = 4312$ mg/L, $[H_2O_2] = 5250$ mg/L, time = 30 min, and pH = 4, while Riaño et al. [54] achieved a removal of 78% in 30 min using $[Fe^{2+}] = 100$ mg/L, $[H_2O_2] = 8000$ mg/L, and pH = 3 for the treatment of a pre-treated SW with 769 mg/L of initial COD. Using other AOPs for the treatment of SW, Garcia et al. [4] obtained COD removals of 45–85.6% by photocatalysis with $TiO_2$ P25, depending on the photocatalyst concentration, irradiation time, and wastewater concentration. Moreover, using another approach, Chen et al. [1] obtained 50.6% COD removal in 35 min by electrocoagulation, while Gomes et al. [55] obtained a COD decrease of 40% by coagulation and a 51% decrease in 4 h by biofiltration with the Asian clam *Corbicula fluminea*. They also showed that the COD could be naturally removed just by bubbling the wastewater with air during a large period of time, due to the aerobic digestion of organic matter by the organisms present in the wastewater.

In a previous study, Domingues et al. [35] conducted an optimization to obtain the best conditions for SW pollutant removal by adsorption using red mud (RM—a waste product from the alumina production industry) and the heterogeneous Fenton at batch conditions with iron filings (IF—a waste from steel production). The single RM adsorption removed up to 55% of COD, while the coagulation with the PDADMAC decreased the COD by 30%. The batch Fenton with the IF reduced the initial COD value by 66% when applied after coagulation, and 86% when applied after the RM adsorption. The best conditions for the adsorption with the RM were using 2.5 g/L of RM during 5 min at pH = 3, while for the Fenton experiments an IF load of 15 g/L and $[H_2O_2] = 50$ mg/L led to the best results. This shows a decrease in the amount of hydrogen peroxide when compared to the studies of Lee et al. [53] and Riaño et al. [54] with similar COD decreases (comparing to the 86% obtained by RM adsorption and Fenton). Therefore, the adsorption with the RM followed by the heterogeneous Fenton method with IF can be an interesting option for the treatment of SW wastewater. Moreover, this can also benefit from the fact that the iron sludge formation is usually more problematic in the homogeneous Fenton than in the heterogeneous form, due to the higher concentration of dissolved $Fe^{2+}$.

In this work, the main objective was to investigate the use of waste materials in the treatment of real swine wastewater, targeting a circular economy approach. For this, adsorption with red mud was evaluated as a potential substitute for PDADMAC coagulation as a pre-treatment. Then, the efficiency of iron fillings waste as the iron source in the Fenton process was investigated. The Fenton process was effectively applied in continuous operation mode using a very low concentration of $H_2O_2$, which should motivate the industrial application of such technology. To the best of our knowledge, no other work reports the treatment of real swine wastewater in continuous operation using RM and IF as raw materials.

## 2. Materials and Methods

### 2.1. Materials and Wastewater Preparation and Characterization

In this work, two different materials were used for the SW treatment. The methodologies for the preparation and characterization of the IF, RM, and SW are presented elsewhere [35,56].

In general, the IF was obtained by disassembling iron wastes from the construction industry, while the RM was provided by an aluminum oxide industry located in Greece, and it was sieved (0.105 mm) before being used. The RM presents a red color, a $pH_{zpc}$ (pH zero point of charge) of 13 [56], a surface area of 0.6 $m^2/g$ [56], and is constituted by several metals (Cu, Zn, Fe, Cr, Ni, Mn, Pb, Mg and Al), with the Fe and Al being the most prevalent ones. The IF presents a specific surface area of 1.14 $m^2/g$ and an average pore diameter of 4.43 nm, and it is practically only composed of iron [57].

The SW was collected from a pig farm located in the central region of Portugal and subsequently diluted with distilled water (8 vol%) to simulate the wastewater resulting from pig farm washing. The SW presented a chemical oxygen demand (COD) of 1700 $mgO_2/L$, biochemical oxygen demand at day 5 ($BOD_5$) of 219 $mgO_2/L$, total solids (TS), Kjeldahl nitrogen (TKN), and phosphorous levels of 1706 mg/L, 245 mg/L, and 12 mg/L, respectively. In addition, it had a pH of 7.5 and a zeta potential of $-19$ mV (pH = 7) and $-10$ mV (at pH = 3) [35].

### 2.2. Coagulation/Adsorption and Heterogeneous Continuous Fenton

Two different approaches were studied for the wastewater treatment. The first employed a coagulation step, and for this, SW was coagulated using a certain volume of the commercial coagulant PDADMAC (0.1%) with a mixing time of 3 min at fast stirring (140 rpm) and 15 min at slow stirring (35 rpm), followed by a sedimentation period of 1 h. A jar-test equipment was used for the coagulation procedure. The second approach implied an adsorption process with RM, based on its good performance in a previous work [35]. For the adsorption process, a load of 2.5 g/L of RM was used with a 5 min contact time, followed by 1 h of sedimentation time, at pH = 3.

For both procedures, after the sedimentation period, the supernatant was separated, a pH adjustment was carried out (pH = 3), and $H_2O_2$ (50 mg/L) was added to the pre-treated wastewater and magnetically stirred. To start the Fenton reaction, a peristaltic pump was used to pump the pre-treated wastewater to a glass column ($d_{internal}$ = 3.1 cm, h = 47 cm) filled with the desired mass of IF and glass spheres filing (GSF) (these spheres only provided porosity to the bulk). Samples were taken at certain times for evaluation, and the wastewater pH was maintained at pH = 3. The COD degradation results are indicated as function of time, divided by the residence time. This was calculated by dividing the sampling time by the residence time of the reactor. The experiments were made at least in duplicate, and the standard deviations are presented in all the figures as error bars.

The $H_2O_2$ concentration in the pre-treated wastewater was controlled over time using indicator strips (0–100 mg/L). The pH was adjusted by using $H_2SO_4$ (6 M) or NaOH (10 M), and the $H_2O_2$ (30% *w/w*) was acquired from Panreac. The batch Fenton experiments were carried out using the SW coagulated with the PDADMAC [35], and, to stop the reactions, the pH was increased to 11. Afterwards, the liquid was separated from the iron filings and the iron sludge, and the cups were left opened in contact with air to allow the $H_2O_2$ decomposition. The experimental conditions (pH, [$H_2O_2$], [IF], RM load, adsorption time, etc.) were selected based on the optimization of the process carried out in our previous batch work [35].

### 2.3. Toxicity Assessment

The toxicity towards *Lepidium sativum* seeds was determined using the Trautmann and Krasny [58] classification for the germination index (GI) method. This phytotoxicity method involves placing 10 seeds in a petri dish and adding 5 mL of each sample (experiments carried out in duplicate with pH adjustment to 6.5–7.5), with an incubation period of 48 h at 27 °C (in dark conditions). As control, ultrapure water was used. The GI was calculated using the relative seed germination (RSG) and the radicle growth (RRG), according to the following equations:

$$\text{RSG (\%)} = \frac{N_G}{N_{G,B}} \times 100 \tag{8}$$

$$\text{RRG (\%)} = \frac{L_R}{L_{R,B}} \times 100 \tag{9}$$

$$\text{GI (\%)} = \frac{\text{RSG} \times \text{RRG}}{100} \tag{10}$$

The $N_G$ and $N_{G,B}$ are the number of germinated seed in the samples and blank, and the $L_R$ and $L_{R,B}$ are the radicle length of the germinated seeds, for samples and blank, respectively.

*2.4. Analytical Methods*

Chemical oxygen demand (COD) was analyzed considering the standard method 5220D [59], using potassium hydrogen phthalate (KHP) for preparing the calibration curves. The digestion occurred at 150 °C for 2 h (ECO25, Velp Scientifica, Usmate Velate, Italy). After that, the samples were left to cool in the dark, at room temperature for 1 h, and their absorbance was measured using a photometer (610 nm) (HI83399 COD photometer, Hanna Instruments, Salaj, Romania). Biochemical oxygen demand ($BOD_5$) was determined by a respirometry measurement, in which the pressure from a closed vessel (20 °C) equipped with a measuring head (OxiTop®-C, WTW, Troistedt, Germany) was monitored, measured, registered, and processed for 5 days by a controller (OxiTop® OC100, WTW). The Kjeldahl nitrogen (TKN) was determined by the Kjeldahl method (DIN EN 13342 [60]), using digestion (DKL, Velp Scientifica) and distillation (UDK, Velp Scientifica) equipment. The phosphorous was determined according to the EPA method 365.3 [61], in which the samples were digested and the absorbance was measured by UV-Vis spectrophotometry at 605 nm (T60, PG instruments Ltd., Lutterworth, UK). The presence of residual iron in the treated solution was determined by flame atomic absorption spectroscopy (ContrAA300, Analytik Jena AG, Jena, Germany).

## 3. Results

*3.1. Batch Fenton Optimization*

Before the continuous Fenton evaluation, a different optimization approach for the batch Fenton was carried out. This approach consisted on the following: (i) evaluation of the pH effect and $[H_2O_2]$ interaction (to understand if the natural pH could be used without a significant loss of performance) and (ii) partial addition of $H_2O_2$ after the 60 min reaction time (to understand if the Fenton reaction was limited by the $H_2O_2$ consumption). For this evaluation, the wastewater was firstly pre-treated with the common coagulation treatment using PDADMAC, and the experiments occurred in plastic reactors placed in a rotatory shaker at 13 rpm. The Fe:$H_2O_2$ molar ratio and the COD content ($mgO_2$/L) divided by the Fe:$H_2O_2$ molar ratio are presented.

Several works report that the $H_2O_2$ and the $Fe^{2+}$ concentrations are the variables with the greatest influence on Fenton efficiency [35,41,45,62–64]. In fact, the best concentrations for $Fe^{2+}$ and $H_2O_2$ are not the same for all the works, which suggests that these parameters should always be optimized for every case. As an example, some of the best $Fe^{2+}$ and $H_2O_2$ concentrations reported for SW treatment are, respectively, 4743 and 5780 mg/L [53], 100 and 400 mg/L [54], or 750 and 750 mg/L [62]. The amount of metal catalyst and oxidant is dependent on the type of wastewater, organic load and characteristics, pre-treatments, apparatus, etc., which, once again, supports the idea that every case should be optimized.

Additionally, it has been shown and agreed upon by different studies that the Fenton reaction presents higher performance at acidic pH (3–4) due to the higher $Fe^{2+}$ solubility [34,57,64]. At a neutral pH range, it is also possible to use the Fenton reaction [34,64], but the performance is lower due to the presence of less reactive radicals (such as •$HO_2$ radicals) and the low solubility of the iron source [65], which precipitates in the $Fe(OH)_3$ form. This is a limitation of traditional Fenton technology. Therefore, the pH effect is also an important parameter that cannot be disregarded.

In a previous work, it was shown that by using the natural wastewater pH, the Fenton efficiency decreases [35]. However, this conclusion can be different if other $H_2O_2$ dosages are considered, as reported in Figure 1. The Fe:$H_2O_2$ molar ratios are 183 and 18, and the COD ($mgO_2$/L)/Fe:$H_2O_2$ molar ratios are 4.6 and 47.2 for $[H_2O_2]$ of 50 mg/L and 500 mg/L, respectively.

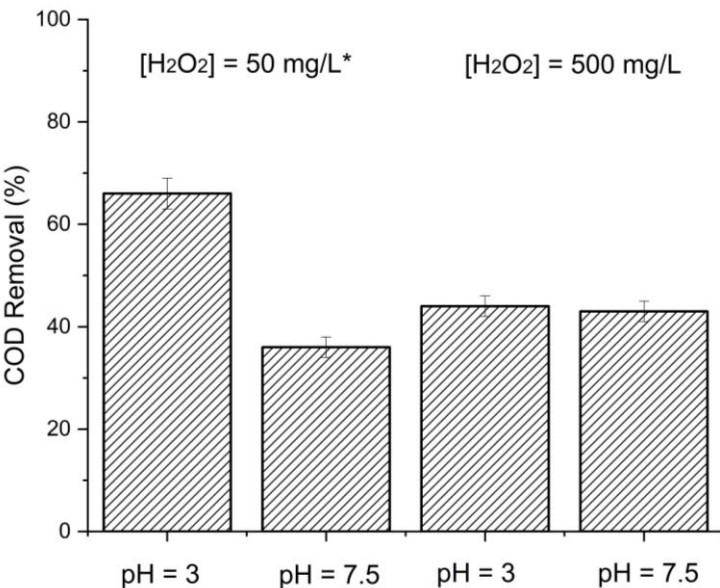

**Figure 1.** Evaluation of pH effect in Fenton reaction (coagulated effluent, [IF] = 15 g/L, t = 60 min, Fe:$H_2O_2$ molar ratio and COD/Fe:$H_2O_2$ ratio: 183 and 4.6 ([$H_2O_2$] = 50 mg/L) and 18 and 47.2 ([$H_2O_2$] = 500 mg/L)). * Experiments previously reported in [35].

In Figure 1, the COD removal is reported for different pH levels, considering different $H_2O_2$ concentrations. It is possible to see that a 10-time increase in the $H_2O_2$ concentration (500 mg/L) decreases the COD removal efficiency (due to a scavenging effect caused by an excessive $H_2O_2$ in the reaction medium), but, when acidic and neutral pH levels are compared using the higher amount of hydrogen peroxide, the results obtained for both pH levels are the same.

In fact, when [$H_2O_2$] = 50 mg/L was considered, the COD removal decreased from 66% to 36% with the increase in the pH from 3 to 7.5, but when a concentration of 500 mg/L was evaluated, the pH = 3 and the pH = 7.5 presented a COD removal of 44% and 43%, respectively. These results suggest that incorrect optimization can lead to similar results between acidic and neutral pH, and it may lead to wrong conclusions. These results may be justified in two ways: (i) at pH = 3 and [$H_2O_2$] = 500 mg/L, the Fenton reaction presents higher efficiency due to the higher solubility of Fe, but COD removal is inhibited due to the scavenging effect caused by excessive $H_2O_2$, which causes low efficiency; and (ii) at pH = 7.5, when [$H_2O_2$] is 50 mg/L, it is completely consumed by self-decomposition, leading to low radical generation and reducing COD removal efficiency. However, when [$H_2O_2$] is 500 mg/L, $H_2O_2$ is in excess, and lower reactive radicals are probably produced [35].

In the industrial application, the $H_2O_2$, the iron source, and pH adjustments can present significant operational costs, which should be decreased and avoided to achieve an economically friendly wastewater treatment. Therefore, these results confirm the need for performing effective optimization, considering a wide range of [$Fe^{2+}$] and [$H_2O_2$], although it has been more than proven that, when good optimization is carried out (as suggested by the results from Figure 1), a pH range of 3–4 is the best range for the Fenton reaction, and such range should be chosen.

Experiments involving additional $H_2O_2$ dosing along the reaction time were performed. After the first Fenton reaction ([IF] = 15 g/L, [$H_2O_2$] = 50 mg/L, pH = 3, time = 60 min), another dose of $H_2O_2$ was added (25, 50, or 500 mg/L), and the reaction continued for an additional 60 min. The COD removal did not substantially change after the additional dose, which suggests that after the initial 60 min, there are still traces of $H_2O_2$, and that the reaction is not limited by the $H_2O_2$ consumption. In fact, Le et al. [66] were able to increase the degradation by only 10% by adding partial $H_2O_2$, which shows that partial $H_2O_2$ dosing may not be an efficient way of increasing the organic contaminant's degradation.

### 3.2. Continuous Fenton Reaction

In a previous study [35], the SW was treated using a pre-treatment (adsorption with RM or volcanic rocks or coagulation with PDADMAC), followed by a batch Fenton reaction with IF. The RM was revealed to be the best material as an adsorbent, so the RM and the IF load optimization was performed. Therefore, this methodology is now evaluated for the treatment of SW in continuous operation.

### 3.2.1. Evaluation of Residence Time

Before the continuous operation, the best operational residence time must be determined to understand the required duration for the efficient reaction to occur in terms of COD removal. Due to this and considering the discontinuous Fenton operation and apparatus [35], several reaction times were evaluated (10, 15, 30, 45, 60, and 90 min). The results are present in Figure 2.

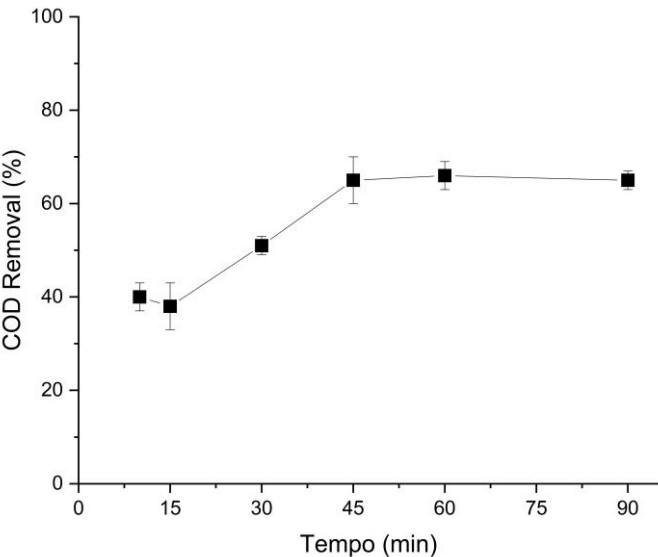

**Figure 2.** Evaluation of time in batch Fenton reaction ([IF] = 15 g/L, [$H_2O_2$] = 50 mg/L, pH = 3, coagulated effluent).

Considering Figure 2, it is possible to see that the best reaction time goes from 45 to 90 min, which gives an extended time range for the reaction to occur inside the column in continuous Fenton. From 10 to 45 min, the COD removal was worse due to the incomplete degradation of organic contaminants. After that treatment time, the removal reached a plateau with no further relevant COD abatement.

In fact, Le et al. [66] evaluated a Fenton reaction for 5 h, and it was observed that the contaminant degradation was the same from 3 to 5 h. The Fenton reaction is usually quick due to a fast $H_2O_2$ consumption [66], and it seems that from 45 to 90 min, the COD removal is inhibited (at the maximum removal), probably due to a higher consumption of $H_2O_2$ in the first 45 min. In fact, this inhibition may be attributed to the depletion of $H_2O_2$ and •OH [42].

### 3.2.2. Continuous Fenton Reaction Assessment

In the continuous Fenton reaction, the mass of IF to be added was 250 g, additionally using another 250 g of GSF to increase the porosity of the bulk. The GSF was also employed to avoid clogging problems in the tubes and in the column, and to increase the turbulence inside the column.

Based on the residence time obtained in Figure 2, a flow rate of 2 mL/min was selected, allowing us to obtain a residence time ($\tau$) of 49 ± 2 min. The results (relative and global COD removal) for continuous Fenton treatment of a pre-treated SW are presented

in Figure 3. The pre-treatment occurred by coagulation with the commercial coagulant PDADMAC or by RM adsorption.

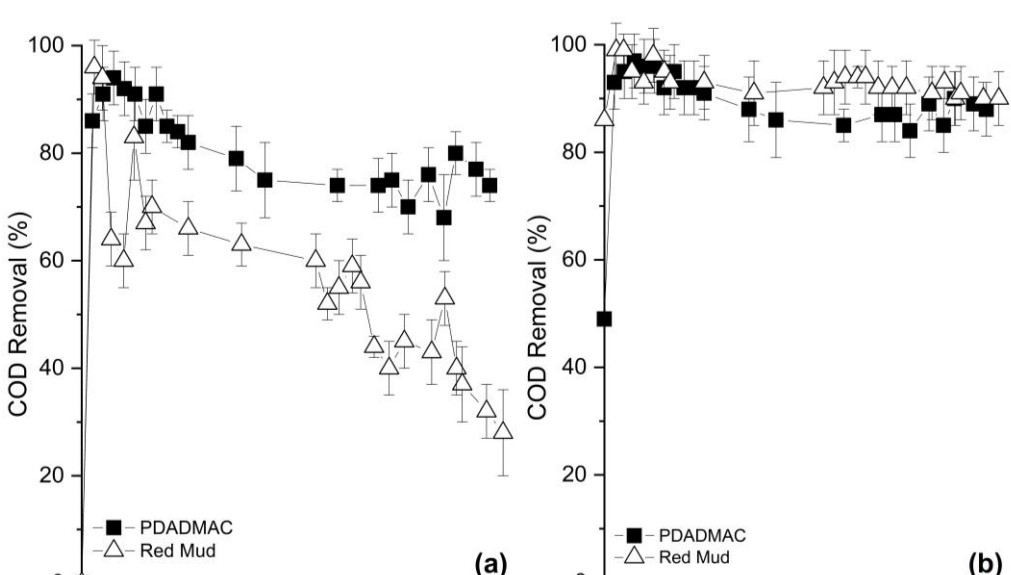

**Figure 3.** (**a**) Relative COD removal during Fenton process after coagulation or adsorption pre-treatment and (**b**) global COD removal with the combined processes (Fenton reaction experimental conditions: pH = 3, $\tau$ = 49 $\pm$ 2 min, mIF = 250 g, mGSF = 250 g. X axis: dimensionless).

The relative COD removal was determined based on the COD of the wastewater after the pre-treatment (850 and 240 mgO$_2$/L for the PDADMAC coagulation and the RM adsorption, respectively) (Figure 3a), while the global COD removal was calculated based on the initial swine wastewater COD value (1700 mgO$_2$/L) (Figure 3b). In Figure 3, the horizontal axis is represented as a function of time/residence time because the samples were not taken at the same time in both reactions, since the process lasted for more than a day in a row. The general results regarding the continuous Fenton treatment are presented in Table 1.

**Table 1.** Characterization of treated wastewater after the different processes.

| Parameter | Initial Value | Fenton after PDADMAC Coagulation | Fenton after RM Adsorption |
|---|---|---|---|
| COD (mgO$_2$/L) | 1700 | 49–264 (850)[1] | 10–171 (240)[1] |
| BOD$_5$ (mgO$_2$/L) | 219 | 4.4 | 30.9 |
| TKN (mg/L) | 1706 | 87.6 | 175.1 |
| Phosphorous (mg/L) | 245 | 0.0 | 0.3 |
| Total Iron (mg/L) | 12 | 58.5 | 55.3 |
| *L. sativum* GI at 48 h | 54.9/Strong inhibition | 92/No inhibition (47.5/Strong inhibition)[2] | 88.7/No inhibition (95/No inhibition)[2] |

Note: [1] COD values and [2] GI results obtained only by the application of pre-treatment (PDADMAC coagulation/RM adsorption).

In the first step, the diluted SW was subjected to a coagulation process with PDAD-MAC [62] or an adsorption treatment with RM as an alternative (load of 2.5 g/L during 5 min at pH = 3). Both pre-treatments followed a sedimentation time of 60 min and only the supernatant was feed to the continuous Fenton reactor. The objectives of this study were as follows: (i) remove the solids present in the SW, (ii) find an efficient substitute of commercial PDADMAC, and (iii) give new life to another industrial residue (RM).

Figure 3a presents the COD abatement profile of the Fenton process after coagulation (with PDADMAC) and adsorption (with RM), based on the COD values determined after the two different pre-treatments (relative COD). The adsorption with RM allowed a decrease of 86% (from 1700 to 240 $mgO_2/L$), while the coagulation with PDADMAC had a decrease of about 50% (from 1700 to 850 $mgO_2/L$), which shows that the adsorption with RM was more effective than the coagulation with the commercial coagulant. When both pre-treated wastewaters were used in the continuous Fenton process, the RM presented higher COD removal (96%), but it rapidly decreased to a COD removal of 64–83%, and as time passed, the COD removal finished at 28%. Analyzing the behavior of the wastewater pre-treated by coagulation with PDADMAC, the relative COD removal ranged between 68–94%, maintaining the COD removal profile more constant (with the last point achieving 74% COD removal) compared to the RM pre-treatment.

These values suggest that the efficiency of COD abatement for the Fenton process, after adsorption with RM, is lower than after coagulation with PDADMAC. However, this is not true if one considers the global COD removal. In fact, although the Fenton efficiency is lower after the RM adsorption when compared to coagulation using PDADMAC, the global amount of COD removal is higher when RM is used as the adsorbent in the pre-treatment. During the Fenton experiment, the highest and lowest COD obtained was 264 and 49 $mgO_2/L$ for PDADMAC and 171 and 10 $mgO_2/L$ for RM, respectively (Table 1). This difference is related to a loss of efficiency that occurs after a certain operational time, linked to catalyst oxidation and activity decay.

Therefore, to make it clear, the global COD removal efficiency is presented in Figure 3b. It is possible to see that combining RM adsorption with the Fenton process led to a higher total COD removal when compared to coagulation with PDADMAC followed by the Fenton process. However, analyzing the higher and lower COD obtained with the RM pre-treatment, it is possible to see that even the samples with higher COD were near the limit imposed by the Portuguese law of 150 $mgO_2/L$ [67] for wastewater discharge in natural courses, while for the PDADMAC pre-treatment, the higher COD obtained was far from this limit. However, both processes could obtain samples bellow the COD limit imposed by law for discharge into certain natural resources, which shows the efficiency of the Fenton treatment in continuous operation.

Analyzing other parameters, both experiments present similar values of leached iron (58.5 and 55.3 mg/L with PDADMAC and RM, respectively) for the last sample, which is related to the oxidation of zero valent iron from the IF. Although the RM pre-treatment was more efficient in removing COD, the coagulation with PDADMAC was more effective in removing $BOD_5$, total Kjeldahl nitrogen, and even in removing organic phosphorous as it can be seen in Table 1.

According to the Portuguese decree of law DL 236/98 [67] the COD, $BOD_5$ (20 °C), total N, total P, and total Fe for wastewater discharge are 150 and 40 $mgO_2/L$, and 15, 10, and 2 mg/L, respectively, and the Portuguese decree of law DL 119/2019 [68], which defines water quality standards for reuse in irrigation, the permissible levels for $BOD_5$, total nitrogen, total phosphorous, and Fe are ≤10–40 $mgO_2/L$ (depending on the irrigation use), 15 mgN/L, 5 mgP/L, and 2 mgFe/L, respectively. Comparing the results from Table 1 and analyzing the organic matter requirements for wastewater discharge, all the experimental conditions led to samples with COD and $BOD_5$ values within the legal limits. Regarding the legal requirements for water reuse for irrigation, the $BOD_5$ maximum level is always respected by the water pre-treated with PDADMAC, while the reclaimed water pre-treated using RM adsorption only complies with the requirements for some irrigation cases. Moreover, the phosphorous requirement also seems to be respected, but unfortunately, the nitrogen and the iron exceed the limit. With this, nitrogen and iron should be removed subsequently so that the water can be used for irrigation. Among the possibilities, the subsequent biological treatment could be a solution to decrease these values, and the Fe could be recovered via ion-exchange or removed by coagulation or biofiltration with the Asian clam [62].

3.2.3. Toxicity Assessment

Several residues from pig farms are used in agriculture as fertilizers [21], and to mitigate water scarcity, wastewater reuse for irrigation in agriculture is the most established end use for reclaimed water [25]. Therefore, to simulate such a strategy of reuse, the toxicity assessment was carried out considering the *L. sativum* seeds' germination index (Table 1).

The $H_2O_2$ concentration was measured after the treatment, and no residual $H_2O_2$ was found, which means that this reactant did not interfere in the toxicity measurements. Considering the Trautmann and Krasny [58] evaluation, the initial wastewater led to a GI of 55%, which is considered to cause a "strong inhibition" in the seed's development. Interestingly, when the pre-treatments were carried out, the RM adsorption led to a wastewater providing a GI of 95%, which is classified as "non-toxic", while the pre-treatment with the PDADMAC coagulation increased the wastewater toxicity, with a GI of about 48%, which was also considered to cause a "strong inhibition". In fact, the results involving RM adsorption were expected to decrease the toxicity due to the efficiency in the removal of organic contaminants, while the results regarding PDADMAC coagulation had the contrary observations. Although the existence of COD decreases due to the coagulation process, the increase in toxicity associated with the use of PDADMAC can be related to the chemical composition of this synthetic coagulant. In fact, the safety data sheet (SDS) for this product reports a lethal concentration of 50% ($LC_{50}$) of 0.74 mg/L at 96 h and 0.23 mg/L at 48 for *Oncorhynchus mykiss* (rainbow trout) and *Daphnia magna* (water flea), respectively. Moreover, Gomes et al. [69] showed that the PDADMAC was the most efficient biocide against the Asian clam, with 100% mortality results. Therefore, the use of this commercial coagulant can present a disadvantage for wastewater treatment due an increase in toxicity, even more so if a biological treatment is to be carried out afterward, since microorganisms can be affected by toxic contaminants present in water.

Analyzing the results after the Fenton experiments, the GI increased for the coagulated samples, being now classified as "non-toxic", and presenting a GI of 92%, which is related to the good performance of Fenton in removing organic contaminants. When the RM pre-treated wastewater was considered, the samples did not show toxicity, although a negligible GI decrease from 95% to 89% was observed. This shows that the toxicity did not increase, so it can be assumed that no relevant toxic by-products were produced during the Fenton process application.

**4. Conclusions**

This work shows a potential usage of waste materials in the wastewater treatment process, which improves sustainable development, following the circular economy approach.

Different approaches to improve the Fenton reaction using coagulated SW were considered. Increasing the $H_2O_2$ concentration to 500 mg/L resulted in the same COD removal using either acidic or neutral pH, which is probably related to the generation of less oxidative radicals (such as $\bullet HO_2$). Moreover, conducting a partial addition of $H_2O_2$ after a 60 min reaction did not improve the COD reduction. This was expected, since before the $H_2O_2$ addition, there were still traces of $H_2O_2$, suggesting that the COD removal was not limited by the total consumption of this reagent.

Considering other SW treatment methods, the adsorption with RM revealed a potential alternative to the coagulation with PDADMAC, being capable of achieving higher COD removal and causing lower toxicity. Moreover, the IF was also revealed to be an attractive source of iron catalyst for the Fenton process in continuous operation. Nevertheless, despite the good COD removal observed with the studied pre-treatments (86% and 50% for the RM adsorption and PDADMAC coagulation, respectively), the COD discharge limits imposed by law could only be respected through the subsequent application of the Fenton reaction. After the overall treatment, the toxicity evaluated by the *L. sativum* GI changed from a "strong inhibition" to a "non-inhibition". Considering other discharge aspects imposed by Portuguese decrees of law, the overall treatment complies with some of the imposed limits, and the application of subsequent treatments (namely, biological and iron removal

processes) should allow for compliance with other mandatory values imposed by the regulations regarding wastewater discharge or water reuse of irrigation.

Therefore, the advantages of this methodology are the efficient use of wastes as raw materials (in which the RM adsorption can efficiently replace the PDADMAC coagulation), the reduction of treatment costs and the contribution to the circular economy (due to the reuse of wastes for a new purpose), as well as the provision of high efficiency for the treatment of real wastewater. As drawbacks, the iron filings must be replaced after some operational time, and other processes will be required to complete the treatment (e.g., remove iron and ammonium/ammonia), for which biological treatments could be a suitable solution for this purpose.

For future work, different types of real wastewater should be tested with this methodology, and scale-up studies should be carried out. The use of photo-Fenton, instead of Fenton, namely using solar radiation, could also be an interesting approach and should not be dismissed. In addition, toxicity studies involving other trophic level species should be carried out.

**Author Contributions:** Conceptualization, J.L., J.G., R.C.M. and E.D.; data curation, J.L. and E.D.; investigation, J.L. and E.D.; methodology, J.L., J.G., R.C.M. and E.D.; visualization, J.L.; writing—original draft preparation, J.L.; writing—review and editing, J.G., R.C.M. and E.D.; supervision J.G., R.C.M. and E.D. All authors have read and agreed to the published version of the manuscript.

**Funding:** This research was funded by Foundation of Science and Technology—FCT (Portugal) grant numbers 2021.06221.BD, CEECIND/01207/2018 and UIDB/00102/2020.

**Data Availability Statement:** The data presented in this study are available on request from the corresponding author.

**Acknowledgments:** The authors gratefully acknowledge the Foundation of Science and Technology—FCT (Portugal).

**Conflicts of Interest:** The authors declare that they have no competing financial interests or personal relationships that could have appeared to influence the work reported in this paper.

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
