# Peer review of "Continuous Heterogeneous Fenton for Swine Wastewater Treatment: Converting an Industry Waste into a Wastewater Treatment Material"

_water, doi:10.3390/w16050781_

Round 1

Reviewer 1 Report

Comments and Suggestions for Authors

Review Report

·       Manuscript ID: water-2878866

·       Title: Continuous heterogenous Fenton for swine wastewater treatment: Converting an industry waste into a wastewater treatment material

·       Authors: João Lincho, João Gomes, Rui C. Martins  and Eva Domingues

·       Section: Wastewater Treatment and Reuse

·       Special Issue: Advanced Processes for Industrial Wastewater Treatment

A.   General Comments:

In this manuscript, swine wastewater (SW) was pre-treated using industrial waste as raw material followed by a continuous heterogenous Fenton process. Before continuous treatment, preliminary bach expriments were also conducted (partly in this manuscript, partly in previous work). In separate experiments, H2O2 concentrations were elevated and different pH values were examined resulting in higher COD removal efficiencies at acidic (=3) pH than as neutral (=7.5) pH, especially at lower H2O2 concentrations. It was concluded that since at acidic pH, Fe (iron catalyst) solubility is higher, H2O2 was more efficiently consumed for oxidation. In addition to that, it is already well-known that excessive H2O2 concentrations inhibit advanced oxidation due to free radical scavenging effects of unreacted, residual H2O2, acting as a free radical scavenger. The strategy of partial, step-wise addition of H2O2 is typically not very useful and the authors re-confirm this fact in their on experimental work. By using iron filings (IF) as the iron catalyst source together with 50 mg/L H2O2, continuous Fenton treatment was applied after pre-treatment using red mud (RM) as the adsorbent or with coagulation with Poly-diallyldimethylammonium chloride (PDADMAC), with RM adsorption resulting in higher COD removal than coagulation. Due to concerns of residual soluble Fe (measured as 55-58 mg/L), toxcity tests were also conducted indicating that the treated effluent sample was not toxic towards Lepidium sativum (a plant from the Brassicaceae family) and could be classified as “non-toxic”, whereas the original and coagulated effluents were inhibitory towards the test organism.

On the other hand, the weakness of this work is that the Fenton optimization approach and studying the effect of multiple/continuous H2O2 additon are not new contributions to the research subject. In fact, the effect of pH and H2O2 on COD remmoval from industrial wastewater are already well-known and studied factors. The originality apparently comes from using waste materials as chemical reagents (for instance, as the heterogeneous Fe catalyst) for industrial effluent treatment. However, characterization of the materials, coupling the Fenton’s reagent with coagulation or adsorption and optimization of the coupled processes have already been reported several times in former studies and also in the previous work of the authors. So, it is important to emphasize the originality and its scientific contribution to the research area in the present work. 

Regarding the toxicity part of the study as well as COD measurements, there is another, analytical prblem. It should also be noted that residual H2O2 concentrations are much more toxic/inhibitory than Fe to most test organisms used in bioassays. The presence of H2O2 also positively interferes with COD test results.

B.    Specific Comments:

Introduction:

·       Instead of the phrase “Moreover, this livestock production is also guilty of high-water consumption….”, the following could be used; “Moreover, this livestock production results in high-water consumption….”.

·       Besides, in the Introduction Section, the focus is mainly put on the presence and release of antibiotics as well as antibiotic resistance. For example; “Several antibiotics are found in the worldwide rivers …… There is also the risk of increasing antibiotic resistance due to the widespread and extensive use of antibiotics. Therefore, it is important to ensure the proper swine wastewater treatment.” What aboout the other parameters that were mentioned and also threaten health and environment?

·       It should also be pointed out that advanced oxidation processes (AOPs) are not alternatives to biological treatment. The author points out the necessity of proper swine wastewater treatment by mentioning biological treatment methods and that these are not efficient enough. Then, they skip to the applcation of AOPs, that is mainly applied to toxic/refractory wastewater for pre-treatment or post-treatment purposes, since AOPs are typically and generally speaking not regarded as stand-alone-treatment options. In fact, the authors also use the Fenton process after pre-treatment via adsorption and coagulation to increase the initial organic load (here measured and followed as the COD parameter, as per cent relative COD removal).

·       In fact, other metals (transition metals) could also be used in Fenton-like reactions, however, iron is the least armful (toxic) one among these alternatives. In this section, after AOPs, the Fenton’s reagent should be introduced and its advantage over the other well-known and successfully applied AOPs (peroxide/ozone- and UV-driven treatment combinations) should be indicated.

Materials and Methods (Experimental):

·       Characterization of the materials used (origin, size, purity, etc.) and the effluent is important. Some more background information could be separately provided as Supplementary Materials and/or in an Appendix Section.

·       All experimental conditions should be explained and why they were specifically chosen (for instance pH, H2O2 concentration and the Fe sourse concentration) in this way. Some supportive References can also be cited. Please add this information for all reaction conditions (the H2O2 to Fe molar ratio for process optimization). Preferrably work with molar units for H2O2 and Fe so that the molar ratio can also be followed.

·       Analyses; Swine wastewater characterization: Were some specific parameters (targets for AOPs) measured, such as antibitotics, heavy metals, etc.

·       In the following phrase, “The COD degradation results …..This was calculated by dividing the sampling time to(?) the residence time of the reactor.” Please check sentence.

·       Please check the following information “H2O2 (33% w/v) was ….“. Typically, it is given on weight basis (H2O2 solution; 30% or 35% w/w).

·       Section 2.3.Please check if “radicle” or “radical” (growth) is meant.

·       The experimental part does not contain a statistical methodology section. Statistical analysis is specially important for toxicity measurement (data processing).

·       Residual H2O2 affect COD and toxicity tests. In order to eliminate its interference it is not only followed/measured, but also destroyed/removed by using chemicals such as sulfide, thiosulfate and catalysts (manganese dioxide powder, also catalase made from bovine liver, for example) to obtain more sensitive and reliable COD / toxicity data. Please add some information about these treatment/preparation procedures.

Results:

·       Section 3.1. RM and SW characterization: Environmental characterization is done before the experimental study. It is not an experimental study result. So, this section is not necessary and it does not belong to this section. It can be given in the Appendix (Supplementary) or the experimental (characterization of the original wastewater-SW) part of the manuscript.

·       The terms “load”, “dose” and “concentration” should be used correctly. For example, the term concentration should be used if the reagent (here: the oxidant H2O2) is added once and at the beginning of the reaction. Otherwise, these terms should be considered if appropriate or when describing the mode of application for H2O2.

·       As the authors state, the factors/variables affecting the Fenton’s reagent (H2O2 and Fe concentrations, pH, etc.) are already well-known/established from previous related work. Also very important is the initial organic matter content and the Fe (catalyst):H2O2 (oxidant) molar ratio of the reaction solution. Step-ise H2O2 does not have a positive effect on COD removals (oxidation rates), as has already been evidenced in previous related work. These factors have to be carefully considered and can also be seen from Figures 1 and 2 in this work.

·       COD results and removal efficiencies: Residual (unreacted) H2O2 has a positive analytical effect of COD measurements. Therefore, if H2O2 remains in the reaction solution, it has to be removed before COD measurements to prevent it effects. Otherwise, COD removal efficiencies are not correct (they would be misleading). This has to be considered when presenting COD removal results. The (positive) interference of H2O2 and other reagents needs to be eliminated particularly before COD and toxicity analyses.

·       The initial H2O2 concentration and Fe:H2O2 molar ratio are critical for oxidation rates and efficiencies. The stoichiometry has to be critically evaluated when optimizing COD removal performance in industrial wastewater. Previous work has already shown that one (single and high concentration) addition of H2O2 once and at the beginning of the reaction is much more effective than adding the same total concentraton of H2O2 in multipe steps over a period of time (for example, during 60 min) which is “H2O2 dosing”. The reason is the reaction between Fe and H2O2 (activation of Fe catalyst by H2O2 oxidant) and the optimum Fe:H2O2 molar ration for effective free radical chain reacton initiation.

·       All experimental conditions should be indicated in all figure captions.

·       Margins of error should be added to all data shown in the figures.

·       Figure 3: On x axis; please correct “Tempo (min)” to “Time (min)”  

·       Figure 4b): From this figure is it evident that the global (overall) COD removal for SW is not different for both combined treatment processes. Interstingly, adsorption could also be an attractive treatment option (adsorption only, without Fenton’s reagent) since it results in an appreciable COD removal and detoxification (presented later) without a combination.

·       Table 2 is really difficult to follow. The information given here about the followed collective environmental parameters before and after different pretreatment/treatment methods could be converted to a column figure or set of figures. Original conditions could be indicated in the figure captions or in the first column of the figure.

·       It is already known that coagulation (due to the release of soluble metal hydroxo complexes and polymeric materials) can increase effluent toxicity. Therefore, coagulation should only be considered for “pre-treatment” purposes.

·       Toxicity usually correlates with oxidation rate as well (better/higher COD removal would result in more detoxification/a reduction in toxicity).

·       Results in Table 3: A difference (change, decrease) in GI from 92% to 89% after adsorption and Fenton’s reagent is not so significant. It indicates that the toxicity does not re-increase.

·       Tables (1-2-3) could be removad into an Appendix or Supplementary Section. In the main manuscript text, figures should be preferred (for Tables 2-3,not Table 1) since these are much eaiser to follow and interpret (also visually more attractive).

Conclusion:

·       This work demonstrated that industrial residues can be effectively used as materials for industrial wastewater treatment in a continuous mode of operation. It could be shown that IF is an attractive source of Fe catalyst material for Fenton treatment of SW. Moreover, the study revealed that RM is a potential (even superior) adsorbent and alternative to coagulation with PDADMAC. The toxicity of the original and treated SW was also evaluated using L. Sativum as the test plant of the bioassay. Apparently, SW exhibited a “strong inhibition”, but after pretreatment and Fenton treatment it changed/decreased to a “non-inhibitory” effect. Adsorption proved to be more attractive not only in terms of pollutant removal but also in terms of ectoxicological effect as compared to coagulation.

Comments on the Quality of English Language

As mentioned above, English only needs minor revisions language editing, re-phrasing, etc.).

Author Response

Reviewer #1 (RW#1):

  1. General Comments:

RW#1: In this manuscript, swine wastewater (SW) was pre-treated using industrial waste as raw material followed by a continuous heterogenous Fenton process. Before continuous treatment, preliminary batch experiments were also conducted (partly in this manuscript, partly in previous work). In separate experiments, H2O2 concentrations were elevated, and different pH values were examined resulting in higher COD removal efficiencies at acidic (=3) pH than as neutral (=7.5) pH, especially at lower H2O2 concentrations. It was concluded that since at acidic pH, Fe (iron catalyst) solubility is higher, H2O2 was more efficiently consumed for oxidation. In addition to that, it is already well-known that excessive H2O2 concentrations inhibit advanced oxidation due to free radical scavenging effects of unreacted, residual H2O2, acting as a free radical scavenger. The strategy of partial, stepwise addition of H2O2 is typically not very useful and the authors re-confirm this fact in their on experimental work. By using iron filings (IF) as the iron catalyst source together with 50 mg/L H2O2, continuous Fenton treatment was applied after pre-treatment using red mud (RM) as the adsorbent or with coagulation with Poly-diallyldimethylammonium chloride (PDADMAC), with RM adsorption resulting in higher COD removal than coagulation. Due to concerns of residual soluble Fe (measured as 55-58 mg/L), toxicity tests were also conducted indicating that the treated effluent sample was not toxic towards Lepidium sativum (a plant from the Brassicaceae family) and could be classified as “non-toxic”, whereas the original and coagulated effluents were inhibitory towards the test organism.

On the other hand, the weakness of this work is that the Fenton optimization approach and studying the effect of multiple/continuous H2O2 addition are not new contributions to the research subject. In fact, the effect of pH and H2O2 on COD removal from industrial wastewater are already well-known and studied factors. The originality apparently comes from using waste materials as chemical reagents (for instance, as the heterogeneous Fe catalyst) for industrial effluent treatment. However, characterization of the materials, coupling the Fenton’s reagent with coagulation or adsorption and optimization of the coupled processes have already been reported several times in former studies and also in the previous work of the authors. So, it is important to emphasize the originality and its scientific contribution to the research area in the present work. 

Answer: We acknowledge the reviewer opinion as well as the recommendations. The novelty of this work was the application of industrial wastes as raw materials for the treatment of real swine wastewater in the continuous operation mode. The application of continuous operation for Fenton’s process in the degradation of real wastewater is not frequently found in literature. On the other, another relevant innovative feature of this work is the application of iron filings (a waste) as catalyst in such process specially using actual effluent. Therefore, the main aim of this work was to show that it is possible to use with great efficacy the heterogenous Fenton process for the treatment of real wastewater in giving new life to industry wastes. The continuous operation can legitimate and inspire the use of such technology at industrial level, since two important factors for industrial operation are addressed for the first time in this work: i) efficiency in the COD abatement with real wastewater scenarios, ii) easy separation of iron from water (although there is still a considerable part of iron dissolved in water), iii) the reuse of wastes as raw materials working in circular economy approach and iv) the great efficiency in continuous operation that it is usually the type of operation used at real-scale. The data obtained in continuous mode using a real effluent can help boosting the scale-up of this technology to the industrial level.

Moreover, the Fenton process optimization with the partial addition of H2O2 and the pH effect at higher H2O2 amounts aimed to understand if such approach was benefic while targeting the degradation of real wastewater, although the effects of increasing the H2O2 and pH are already well known and reported by several authors, as also explained in our manuscript. Still, most of the works found in literature deal with simulated wastewater and not with real effluents with complex composition that may influence (positively or negatively) the degradation results.

With this, and to enhance this work novelty, the following sentence was added to the manuscript (introduction, lines 177-185):

“In this work, the main objective was to investigate the use of waste materials in the treatment of real swine wastewater, targeting a circular economy approach. For this, adsorption with red mud was evaluated as a potential substitute for PDADMAC coagulation as a pre-treatment. Then, the efficiency of iron fillings waste as the iron source in Fenton’s process was investigated. The Fenton process was effectively applied in continuous operation mode using a very low concentration of H2O2, which should motivate the industrial application of such technology. To the best of our knowledge, no other work reports the treatment of real swine wastewater in continuous operation using RM and IF as raw materials.”.

RW#1: Regarding the toxicity part of the study as well as COD measurements, there is another, analytical problem. It should also be noted that residual H2O2 concentrations are much more toxic/inhibitory than Fe to most test organisms used in bioassays. The presence of H2O2 also positively interferes with COD test results.

Answer: Thank you for your attention. In fact, the addition of H2O2 can severely interfere and increase the COD results and affect the toxicological tests, since the H2O2 can be efficiently used for disinfection.

However, in the present study this was not a problem since the H2O2 applied in the continuous operation was totally consumed due to the high amount of iron inside the column. In fact, the amount of H2O2 in the effluent leaving the reactor was controlled using H2O2 (0-1000 mg/L) test strips and H2O2 concentration was always below the detection limited. Thus, it was ensured that there was no interreference coming from this reactant in the posterior analytical methods.

This information was added to the manuscript (section 3.2.3 – toxicity assessment, lines 446-447):

“The H2O2 concentration was measured after the treatment and no residual H2O2 was found, which means that this reactant did not interfere in the toxicity measurements.”.

  1. Specific Comments:

Introduction:

RW#1: Instead of the phrase “Moreover, this livestock production is also guilty of high-water consumption….”, the following could be used; “Moreover, this livestock production results in high-water consumption….”.

Answer: Thank you for the note. This will be corrected in the manuscript (lines 37-38).

RW#1: Besides, in the Introduction Section, the focus is mainly put on the presence and release of antibiotics as well as antibiotic resistance. For example, “Several antibiotics are found in the worldwide rivers …… There is also the risk of increasing antibiotic resistance due to the widespread and extensive use of antibiotics. Therefore, it is important to ensure the proper swine wastewater treatment.” What about the other parameters that were mentioned and also threaten health and environment?

Answer: We acknowledge your opinion. The authors agree with reviewer and more information regarding to the other aspects evaluated in this work were added to the introduction as mentioned (lines 46-56):

“The natural organic matter (NOM) occurs due to the residues from plants, animals, and humic substances while synthetic organic matter (SOM) is usually complex, dangerous and appears due to the use of chemicals in industries or agriculture [13]. The high organic loads of swine wastewater are a threat to the environment if not properly treated.  Phosphorous and nitrogen are macronutrients that can lead to the eutrophication phenomenon [14,15]. Ammonium is usually present at high concentration in swine wastewater [16], and although it does not present toxic effects, it can cause odors, microbial development [17]. Ammonia can cause acidification of water, eutrophication, or toxicity and other harmful effects on aquatic organisms [18]. Moreover, SW can present high levels of virus and protozoan agents which can be a source of diseases [19], and therefore the disinfection of this wastewater should also be considered.”.

References:

  1. Hashim, K.; Saad, W. I.; Saffa, K.; Al-Janabi, A. Effects of organic matter on the performance of water and wastewater treatment: Electrocoagulation a case study. IOP Conf. Ser. Mater. Sci. Eng. 2021, 1184:012018.
  2. Ye, Y.; Ngo, H. H.; Guo, W.; Liu, Y.; Li, J.; Liu, Y.; Zhang, X.; Jia, H. Insight into chemical phosphate recovery from municipal wastewater. Sci. Total Environ. 2017, 576, 159-171.
  3. Yang, S.; Peng, S.; Xu, J.; He, Y.; Wang, Y. (2015) Effects of water saving irrigation and controlled release nitrogen fertilizer managements on nitrogen losses from paddy fields. PAWE. 2015, 13, 71-80.
  4. Prokhorova, A.; Kainuma, M.; Hiyane, R.; Boerner, S.; Goryanin, I. Concurrent treatment of raw and aerated swine wastewater using an electrotrophic denitrification system. Bioresour. Technol. 2021, 322, 124508.
  5. Radu, G.; Racoviteanu, G. Removing ammonium from water intended for human consumption. A review of existing tech-nologies. IOP Conf. Ser. Earth Environ. Sci., 2021, 664:012029.
  6. Soler, P.; Faria, M.; Barata, C.; García-Galea, E.; Lorente, B.; Vinyoles, D. Improving water quality does not guarantee fish health: Effects of ammonia pollution on the behaviour of wild-caught pre-exposed fish. PLoS ONE. 2021, 16(8).
  7. Garcia, B. B.; Lourinho, G.; Romano, P.; Brito, P. S. D. Photocatalytic degradation of swine wastewater on aqueous TiO2 suspensions: Optimization and modeling via Box-Behnken design. Heliyon, 2020, 6(1).

RW#1:  It should also be pointed out that advanced oxidation processes (AOPs) are not alternatives to biological treatment. The author points out the necessity of proper swine wastewater treatment by mentioning biological treatment methods and that these are not efficient enough. Then, they skip to the application of AOPs, that is mainly applied to toxic/refractory wastewater for pre-treatment or post-treatment purposes, since AOPs are typically and generally speaking not regarded as stand-alone-treatment options. In fact, the authors also use the Fenton process after pre-treatment via adsorption and coagulation to increase the initial organic load (here measured and followed as the COD parameter, as per cent relative COD removal).

Answer: Thank you so much for your important commentary. The advanced oxidation processes are effective in removing toxic and recalcitrant organic contaminants from water, but the biological treatment should not be discarded from any wastewater treatment. In fact, for SW anaerobic digestion followed by aerobic biological processes are usually able to remove a high load of the organic pollutants present in this wastewater. However, the treated effluent is still usually not prone for discharge or for water reuse. Indeed, the increasing presence of toxic and biorefractory molecules in this effluent (such as antibiotics) makes it necessary to have complementary processes able to further depurate the biologically treated effluent. The water regulations are getting stricter and the need for water reclamation claims for the introduction of advanced oxidation processes in the global wastewater treatment train.

For these cases, the application of Fenton’s reaction can be a suitable complement either as pre-treatment to decrease the wastewater toxicity and allow to the biological treatment be carried posteriorly in more effective way or as a post-treatment to refine the water. Agreeing with the reviewer, the following sentence was added to the manuscript (lines 85-90):

“In this perspective, the AOPs can be applied after biological treatment to remove recalcitrant contaminants that the biological route was not capable to, while when used before, they can reduce the wastewater toxicity and improve the forthcoming biological treatment efficiency [28]. Regarding to this, the Fenton’s reaction can be a suitable solution due to its simplicity and high efficiency in the removal of harmful pollutants from water and wastewater.

Reference:

  1. Martins, R. C.; Quinta-Ferreira, R. M. Phenolic wastewaters depuration and biodegradability enhancement by ozone over active catalysts. Desalination. 2011, 270, 90-97.

RW#1: In fact, other metals (transition metals) could also be used in Fenton-like reactions, however, iron is the least harmful (toxic) one among these alternatives. In this section, after AOPs, the Fenton’s reagent should be introduced and its advantage over the other well-known and successfully applied AOPs (peroxide/ozone- and UV-driven treatment combinations) should be indicated.

Answer: We acknowledge your opinion and agree with you. The following introduction to Fenton’s reaction was added to the introduction section (lines 91-147):

“[…]. Fenton’s reaction uses Fe2+ to initiate and catalyze H2O2 decomposition forming hydroxyl radicals (•OH) and Fe3+, and due to the action of H2O2 at the same time, the Fe3+ is reduced to Fe2+ forming the hydroperoxyl radical (•HO2), allowing the chain reaction to continue [29,30]. The Fe2+ and the H2O2 can also be •OH scavengers. The Fe2+ in excess can react with the •OH radicals and form Fe3+ and OH- [30,31], while the H2O2 can cause a low radical generation decreasing the Fenton’s efficiency when is in low amount, and when it is in excess it can produce •HO2 or self-decompose in water and oxygen [32,33]. The key parameters are the concentration of Fe and H2O2, pH, temperature, and concentration of organic and inorganic species. To obtain the maximum performance is also necessary to understand the relation between [Fe2+]/[H2O2] ratio and the •OH production and consumption [34].

(1)

(2)

(3)

(4)

(5)

Typically, the Fenton’s reaction occurs as a homogeneous system [29], and often at low pH with the best reported pH of 3-4 [34,35]. Neutral pH can be used but the efficiency is lower [36], and at basic pH, the Fenton reaction is limited since the iron precipitates in Fe(OH)3 form [37]. Also, at high pH, the hydrogen peroxide can decompose in oxygen and water. Other metals as Cu, Ce, Mn, Cr Co, Ru or Al can be used in processes known as Fenton-like processes [30], and although the Fenton-like reactions can be used in neutral or basic pH, they can present disadvantages as high-costs, metal leaching, complex mechanisms, and low catalyst reuse [30]. The iron sludge production is a drawback of homogeneous Fenton’s, but the use of heterogeneous materials could overcome this issue. Using zero valent iron as iron source can be advantageous since is cheap and widely available (iron powders, filings, wires, nails, wool, or nanoparticles), simple and easy to be handled [38]. The Fe0 can be oxidized to Fe2+ by different mechanisms (Equation 1-4) generating •OH or other radicals in the classic Fenton’s reaction mechanism [38,39]. However, the use of Fe0 has the unique advantage of being capable to regenerate the Fe3+ into Fe2+ at the iron surface (Equation 5), being a cost-saving process when compared to the typical regeneration of Fe3+ by H2O2 action [40,41]

Fenton can present several advantages when compared to other AOPs. It is characterized by low-cost, simplicity and low toxicity of the required materials [42]. When coupled with radiation (photo-Fenton) it can increase the efficiency and decrease the required amount of Fe2+ and H2O2 [35, 43-46]. The presence of radiation favors the reduction of Fe3+ into Fe2+ also forming •OH (Equation 6) [38]. Using the Sun as radiation source can be an interesting alternative to promote photo-Fenton without increasing the energy costs [37, 47-49].

(6)

Regarding to other technologies, ozonation can be efficient in removing color, odor, taste, and pollutants from wastewater, but the action of ozone molecule is selective, and it is associated to low gas/liquid transference rates and to poor COD and TOC removal. Moreover, it can present as drawback the short lifetime of ozone and high energy demand [37, 50]. Combining ozone with hydrogen peroxide (peroxone process) can enhance the reaction kinetics, improving the •OH generation and reducing the amount of ozone required but the reaction initiation step is slow [30,31]. Combining radiation with H2O2 can also be used for •OH generation (Equation 7), but this reaction has a low quantum yield and requires UVC radiation [31,38].

(7)

Photocatalysis requires radiation and a photocatalyst, and it is usually associated to low reaction kinetics and high recombination rates [35,51,52]. Adding H2O2 to photocatalysis allows the generation of additional •OH radicals due to better trapping of conduction band electrons or reaction with superoxide radicals [38]. Unfortunately, the traditional materials used in photocatalysis are in powder forms which implies difficult operations to recover the photocatalyst and is an obstacle to the industrial application [53]. Therefore, Fenton can be a suitable technology over other advanced oxidation process, and by using industrial wastes as iron source, it increases the economic feasibility of Fenton maintaining high efficiency, simplicity, and easy separation of treated liquid from the iron source.”.

References:

  1. Domingues, E.; Gomes, J.; Quina, M.; Quinta-Ferreira, R.; Martins, R. Detoxification of Olive Mill Wastewaters by Fenton’s Process. Catalysts 2018, 8, 662,
  2. Nidheesh, P.V.; Couras, C.; Karim, A.V.; Nadais, H. A Review of Integrated Advanced Oxidation Processes and Biological Processes for Organic Pollutant Removal. Chem Eng Commun 2021, 1–43.
  3. M’Arimi, M.M.; Mecha, C.A.; Kiprop, A.K.; Ramkat, R. Recent Trends in Applications of Advanced Oxidation Processes (AOPs) in Bioenergy Production: Review. Renew. Sust. Energ. Rev. 2020, 121, 109669.
  4. Boczkaj, G.; Fernandes, A. Wastewater Treatment by Means of Advanced Oxidation Processes at Basic pH Conditions: A Review. Chem. Eng. J. 2017, 320, 608–633.
  5. Prieto-Rodríguez, L.; Spasiano, D.; Oller, I.; Fernández-Calderero, I.; Agüera, A.; Malato, S. Solar Photo-Fenton Optimization for the Treatment of MWTP Effluents Containing Emerging Contaminants. Catal. Today 2013, 209, 188–194.
  6. Neyens, E.; Baeyens, J. A Review of Classic Fenton’s Peroxidation as an Advanced Oxidation Technique. J. Hazard. Mater. 2003, 98, 33–50.
  7. Saritha, P.; Aparna, C.; Himabindu, V.; Anjaneyulu, Y. Comparison of Various Advanced Oxidation Processes for the Deg-radation of 4-Chloro-2 Nitrophenol. J. Hazard. Mater. 2007, 149, 609–614.
  8. Domingues, E.; Lincho, J.; Fernandes, M.J.; Gomes, J.; Martins, R.C. Low-Cost Materials for Swine Wastewater Treatment Using Adsorption and Fenton’s Process. Environ. Sci. Pollut. Res. 2023.
  9. Kanakaraju, D.; Glass, B.D.; Oelgemöller, M. Advanced Oxidation Process-Mediated Removal of Pharmaceuticals from Water: A Review. J. Environ. Manage. 2018, 219, 189–207.
  10. Litter, M.; Quici, N. Photochemical Advanced Oxidation Processes for Water and Wastewater Treatment. Recent Pat. Eng. 2010, 4(3), 217-241.
  11. Chang, M.-C-; Shu, H.-Y.; Yu, H.-H. Olive mill wastewater degradation by Fenton oxidation with zero-valent iron and hy-drogen peroxide. J. Hazard. Mater. 2006, B138, 574–581.
  12. Kallel, M.; Belaid, C.; Boussahel, R.; Ksibi, M.; Montiel, A.; Elleuch, B. An integrated technique using zero-valent iron and UV/H2O2 sequential process for complete decolorization and mineralization of C.I. Acid Black 24 wastewater. J. Hazard. Mater. 2009, 163, 550–554.
  13. Bremner, D. H.; Burgess, A. E.; Houllemare, D.; Namkung, K.-C. Phenol degradation using hydroxyl radicals generated from zero-valent iron and hydrogen peroxide. Appl. Catal. B. 2006, 63, 15-19.
  14. Gamarra-Güere, C.D.; Dionisio, D.; Santos, G.O.S.; Vasconcelos Lanza, M.R.; de Jesus Motheo, A. Application of Fenton, Photo-Fenton and Electro-Fenton Processes for the Methylparaben Degradation: A Comparative Study. J. Environ. Chem. Eng. 2022, 10, 106992.
  15. Riga, A.; Soutsas, K.; Ntampegliotis, K.; Karyannis, V.; Papapolymerou, G. Effect of system parameters and of inorganic salts on the decolorization and degradation of Procion H-exl dyes. Comparison of H2O2/UV, Fenton, UV/Fenton, TiO2/UV and TiO2/UV/H2O2 processes. Desalination. 2007, 211, 72-86.
  16. Méndez-Arriaga, F.; Esplugas, S.; Giménez, J. Degradation of the emerging contaminant ibuprofen in water by photo-Fenton. Water Res. 2010, 44, 589-595.
  17. Lopez, N.; plaza, S.; Afkhami, A.; Marco, P.; Giménez, J.; Esplugas, S. Treatment of Diphenhydramine with different AOPs including photo-Fenton at circumneutral pH. J. Chem. Eng. 2017, 318, 112-120.
  18. Funai, D.H.; Didier, F.; Giménez, J.; Esplugas, S.; Marco, P.; Machulek, A. Photo-Fenton Treatment of Valproate under UVC, UVA and Simulated Solar Radiation. J. Hazard. Mater. 2017, 323, 537–549.
  19. Carra, I.; Santos-Juanes, L.; Fernández, G. A.; Malato, S.; Pérez, J. A. S. New approach to solar photo-Fenton operation. Ra-ceway ponds as tertiary treatment technology. J. Hazard. Mater. 2014, 279, 322–329.
  20. Rodríguez, M.; Malato, S.; Pulgarin, C.; Contreras, S.; Curcó, D.; Giménez, J.; Esplugas, S. Optimizing the solar photo-Fenton process in the treatmen of contaminated water. Determination of intrinsic kinetic constants for scale-up. Sol. Energy. 2005, 79, 360-368.
  21. Sánchez-Pérez, J. A.; Soriano-Molina, P.; Rivas, G.; García Sánchez, J. L.; Casas López, J. L.; Fernández Sevilla, J. M. Effect of temperature and photon absorption on the kinetics of micropollutant removal by solar photo-Fenton in raceway pond reactors. J. Chem. Eng. 2017, 310, 464-472.
  22. Xiao, J.; Xie, Y.; Cao, H. Organic pollutants removal in wastewater by heterogeneous photocatalytic ozonation. Chemiosphere. 2015, 121, 1-17.
  23. Pelaez, M., Nolan, N. T., Pillai, S. C., Seery, M. K., Falaras, P., Kontos, A. G., Dunlop, P. S. M., Hamilton, J. W. J., Byrne, J. A., O’Shea, K., Entezari, M. H., Dionysiou, D. D. A review on the visible light active titanium dioxide photocatalysts for envi-ronmental applications. Appl. Catal. B. 2012, 125, 331–349.
  24. Mecha, A. C.; Chollom, N. N. Photocatalytic ozonation of wastewater: a review. Environ. Chem. Lett. 2020, 18, 1491-1507.
  25. Shan, A. Y., Ghazi, T. I. Mohd., Rashid, S. A. Immobilisation of titanium dioxide onto supporting materials in heterogeneous photocatalysis: A review. Appl. Catal. A-Gen. 2010, 389(1–2), 1–8.

Materials and Methods (Experimental):

RW#1: Characterization of the materials used (origin, size, purity, etc.) and the effluent is important. Some more background information could be separately provided as Supplementary Materials and/or in an Appendix Section.

Answer: Thank you so much for your opinion. Detailed information was already published previously. However, and in accordance with opinion of the reviewer, the following information was added to the section 2.1 (lines 188-203): (Table 1 presenting the SW characterization was deleted).

“In this work, two different materials were used for the SW treatment. The methodologies for the preparation and characterization of the IF, RM and SW are presented elsewhere [36, 57]. 

In general, the IF was obtained by disassembling iron wastes from construction industry, while the RM was provided by an aluminum oxide industry located in Greece and sieved (0.105 mm) before being used. The RM presents a red color, a pHzpc (pH zero point of charge) of 13 [57], a surface area of 0.6 m2/g [57], and constituted by several metals (Cu, Zn, Fe, Cr, Ni, Mn, Pb, Mg and Al) with the Fe and Al being the most prevalent. The IF presented a specific surface area of 1.14 m2/g and an average pore diameter of 4.43 nm, and it is practically only composed by iron [58].

The SW was collected from a pig farm located in the center region of Portugal and posteriorly diluted with distilled water (8 vol%) to simulate the wastewater that is resultant from the pig farm washing. The SW presented a chemical oxygen demand (COD) of 1700 mgO2/L, biochemical oxygen demand at day 5 (BOD5) of 219 mgO2/L, total solids (TS), Kjeldahl nitrogen (TKN), phosphorous of 1706 mg/L, 245 mg/L, and 12 mg/L respectively, pH of 7.5, and a zeta potential of -19 mV (pH = 7) and -10 mV (pH = 3) [36].”.

RW#1: All experimental conditions should be explained and why they were specifically chosen (for instance pH, H2O2 concentration and the Fe source concentration) in this way. Some supportive References can also be cited. Please add this information for all reaction conditions (the H2O2 to Fe molar ratio for process optimization). Preferably work with molar units for H2O2 and Fe so that the molar ratio can also be followed.

Answer: Thank you for your opinion. The experimental conditions (pH, H2O2 and Fe concentration) were chosen based on the optimization carried in our previous work and this information was added to the manuscript (section 2.2, lines 229-231). Moreover, other information regarding to the Fe:H2O2 molar ratio was added to the Figures caption along the manuscript.

“[…]. The experimental conditions (pH, [H2O2], [IF], RM load, adsorption time, etc.) were selected based on the optimization of the process carried out in our previous work in batch [36].”.

RW#1: Analyses; Swine wastewater characterization: Were some specific parameters (targets for AOPs) measured, such as antibiotics, heavy metals, etc.

Answer: Thank you for your opinion. In this work, the COD, and other chemical parameters like, nitrogen, and phosphorous were followed to attest the possibility of reusing the treated water for agriculture irrigation. Currently, these are the only parameters requested by the Directive on the minimum requirements for water reuse. But in future work, it is our aim to assess microcontaminants in the wastewater since it is likely that novel legislation regarding water management will consider also these parameters.

RW#1: In the following phrase, “The COD degradation results ...This was calculated by dividing the sampling time to(?) the residence time of the reactor.” Please check sentence.

Answer: Thank you for this correction. This was a mistake and was revised accordingly (section 2.2, lines 219-221) to: “The COD degradation results are indicated as function of time divided by the residence time. This was calculated by quotient between sampling time and the residence time of the reactor.”.

RW#1: Please check the following information “H2O2 (33% w/v) was ….“. Typically, it is given on weight basis (H2O2 solution; 30% or 35% w/w).

Answer: Thank you for this note. The H2O2 solution is 30% w/w. This information was corrected in the manuscript (section 2.2, line 225).

RW#1: Section 2.3. Please check if “radicle” or “radical” (growth) is meant.

Answer: Thank you for your annotation. This was a mistake and was already corrected. The correct name is “radicle” since it is regarding to the seeds (section 2.3, line 239)..

RW#1: The experimental part does not contain a statistical methodology section. Statistical analysis is specially important for toxicity measurement (data processing).

Answer: Thank you for your opinion. All the toxicity tests were made in duplicate, and no other statistical analysis were made. This information was added to the manuscript (section 2.3).

RW#1: Residual H2O2 affect COD and toxicity tests. In order to eliminate its interference, it is not only followed/measured, but also destroyed/removed by using chemicals such as sulfide, thiosulfate and catalysts (manganese dioxide powder, also catalase made from bovine liver, for example) to obtain more sensitive and reliable COD / toxicity data. Please add some information about these treatment/preparation procedures.

Answer: Thank you for your considerations. Regarding to the batch experiments, after the reaction time, the reaction was stopped by increasing the pH to 11-12 which allows the precipitation of iron as iron hydroxides form (Fe(OH)2 or Fe(OH)3) and the liquid was then separated from the iron sludges and oxidized iron filings. The COD tests were only carried after this and after another certain time to allow to the residual H2O2 to disappear naturally. To help the H2O2 vanishing, the cups were kept open to the air and were agitated several times. When the aim was to evaluate the dissolved iron in the samples, the reaction was not stopped. In that case only the supernatant was separated from iron filings and immediately analyzed by atomic absorption spectroscopy.

For the continuous experiments, the final solution did not have any residual H2O2 (analyzed via H2O2 strips of range 0-100 mg/L), so the COD and the toxicity were posteriorly carried as described in the manuscript. Therefore, it was not used any other agent to eliminate the residual H2O2. Moreover, the use of sulfide or thiosulfate is not desired because it can interfere with the COD measurements.

Therefore, the following information was added to section 2.2 (lines 226-229): “The batch Fenton experiments were carried using the SW coagulated with the PDADMAC [36], and to stop the reactions the pH was increased to 11. Afterwards, the liquid was separated from the iron filings and the iron sludge, and the cups were left opened in contact with air to allow the H2O2 decomposition.

Moreover, at section 3.2.3 (lines 446-447) it was introduced the following sentence: “The H2O2 concentration was measured after the treatment and no residual H2O2 was found, which means that this reactant did not interfere in the toxicity measurements.“.

Results:

RW#1: Section 3.1. RM and SW characterization: Environmental characterization is done before the experimental study. It is not an experimental study result. So, this section is not necessary, and it does not belong to this section. It can be given in the Appendix (Supplementary) or the experimental (characterization of the original wastewater-SW) part of the manuscript.

Answer: We acknowledge your opinion. Changes were made accordingly.

RW#1: The terms “load”, “dose” and “concentration” should be used correctly. For example, the term concentration should be used if the reagent (here: the oxidant H2O2) is added once and at the beginning of the reaction. Otherwise, these terms should be considered if appropriate or when describing the mode of application for H2O2.

Answer: Thank you so much for this observation. Changes were made accordingly in all the manuscript.

RW#1: As the authors state, the factors/variables affecting the Fenton’s reagent (H2O2 and Fe concentrations, pH, etc.) are already well-known/established from previous related work. Also very important is the initial organic matter content and the Fe (catalyst):H2O2 (oxidant) molar ratio of the reaction solution. Stepwise H2O2 does not have a positive effect on COD removals (oxidation rates), as has already been evidenced in previous related work. These factors have to be carefully considered and can also be seen from Figures 1 and 2 in this work.

Answer: Thank you for this consideration. The required information will be added in the manuscript text and figures caption.

RW#1: COD results and removal efficiencies: Residual (unreacted) H2O2 has a positive analytical effect of COD measurements. Therefore, if H2O2 remains in the reaction solution, it has to be removed before COD measurements to prevent it effects. Otherwise, COD removal efficiencies are not correct (they would be misleading). This has to be considered when presenting COD removal results. The (positive) interference of H2O2 and other reagents needs to be eliminated particularly before COD and toxicity analyses.

Answer: Thank you for this consideration. As already mentioned, for the batch experiments the reaction was stopped (increasing the pH to 11), separated from the iron filings, and then it passed a certain time to allow the “natural” vanishment of H2O2. When the continuous experiments were carried, all the H2O2 was consumed during the reaction (due to the high amount of iron present inside the column) and therefore, no residual H2O2 was observed in the samples, after measuring with the H2O2 strips. With this, the COD toxicity was related to the elimination of organic pollutants and the toxicity reduction at the Lepidium sativum was related to the COD removal.

RW#1: The initial H2O2 concentration and Fe:H2O2 molar ratio are critical for oxidation rates and efficiencies. The stoichiometry has to be critically evaluated when optimizing COD removal performance in industrial wastewater. Previous work has already shown that one (single and high concentration) addition of H2O2 once and at the beginning of the reaction is much more effective than adding the same total concentration of H2O2 in multiple steps over a period of time (for example, during 60 min) which is “H2O2 dosing”. The reason is the reaction between Fe and H2O2 (activation of Fe catalyst by H2O2 oxidant) and the optimum Fe:H2O2 molar ration for effective free radical chain reaction initiation.

Answer: We acknowledge your opinion. The required information was added to the manuscript text and to the Figures caption. The section 3.1.2 “H2O2 partial addition effect” was revised due to a misleading in the results, and added to the section 3.1.1, since no substantial changes were seen now.

RW#1: All experimental conditions should be indicated in all figure captions.

Answer: Thank you for your consideration. Changes were made accordingly (Figure 3).

RW#1: Margins of error should be added to all data shown in the figures.

Answer: Thank you for this note. The manuscript was revised accordingly (Figure 3).

RW#1: Figure 3: On x axis; please correct “Tempo (min)” to “Time (min)”  

Answer: Thank you. This was corrected (new Figure 2).

RW#1:  Figure 4b): From this figure is it evident that the global (overall) COD removal for SW is not different for both combined treatment processes. Interestingly, adsorption could also be an attractive treatment option (adsorption only, without Fenton’s reagent) since it results in an appreciable COD removal and detoxification (presented later) without a combination.

Answer: Thank you for this consideration. Although the overall Fenton profiles are similar (with the adsorption using RM presenting a slightly better COD removal), it is important to see that the use of RM adsorption allows to decrease the COD from 1700 to 240 mgO2/L (86% removal) while the PDADMAC causes the reduction to 850 mgO2/L (50% removal). When these processes are coupled to Fenton, the COD obtained with Fenton after the PDADMAC coagulation is between 49-264 while when obtained after the RM adsorption is between 10-171 mgO2/L. So, although the RM adsorption seems to be a good option for being used alone, it cannot achieve the wastewater COD discharge limits (150 mgO2/L) by itself, and this is why the Fenton’s should also be applied. And although not evaluated, the presence of H2O2 should also contribute for the wastewater disinfection in which is also an advantage of using Fenton’s reaction. Regarding to the low toxicity involving the RM, it is definitely related to the high COD removal.

RW#1: Table 2 is really difficult to follow. The information given here about the followed collective environmental parameters before and after different pretreatment/treatment methods could be converted to a column figure or set of figures. Original conditions could be indicated in the figure captions or in the first column of the figure.

Answer: Thank you. The referred table was revised to ensure a reader-friendly information. It was changed from this:

Parameter

PDADMAC coagulation experiment

RM adsorption experiment

Initial wastewater COD (mgO2/L)

1700

Pre-treated wastewater COD (coagulation or adsorption) (mgO2/L)

850

240

Highest COD obtained during Fenton process (mgO2/L)

264

171

Lowest COD obtained during Fenton process (mgO2/L)

49

10

BOD5 (mgO2/L)

4.4

30.9

TKN (mg/L)

87.6

175.1

Phosphorous (mg/L)

0

0.3

Total Iron (mg/L)

58.5

55.3

To this:

Table 1. Characterization of treated wastewater after the different processes.

Parameter

Initial value

Fenton after PDADMAC coagulation

Fenton after RM adsorption

COD (mgO2/L)

1700

49-264

(850)1

10-171

(240)1

BOD5 (mgO2/L)

219

4.4

30.9

TKN (mg/L)

1706

87.6

175.1

Phosphorous (mg/L)

245

0.0

0.3

Total Iron (mg/L)

12

58.5

55.3

L. sativum GI at 48 h

54.9 / Strong inhibition

92 / No inhibition

(47.5 / Strong inhibition)2

88.7 / No inhibition

(95 / No inhibition)2

1 COD values and, 2 GI results obtained only by the application of pre-treatment (PDADMAC coagulation/RM adsorption)

RW#1: It is already known that coagulation (due to the release of soluble metal hydroxy complexes and polymeric materials) can increase effluent toxicity. Therefore, coagulation should only be considered for “pre-treatment” purposes.

Answer: Thank you for this consideration. The coagulation and sedimentation are an important pre-treatment to be applied in this case, due to the high amount of suspended solids in the swine wastewater. The presence of solids in the column would cause clogging problems easily, and therefore, a good pre-treatment must be used. Interestingly, the RM can also be effectively used for this purpose, presenting also higher COD removal than the PDADMAC. Also, the cause of a higher toxicity after the coagulation treatment (in which the COD is lower) is caused due to the problems mentioned by the reviewer. In this case, also the red mud presents an advantage.

RW#1: Toxicity usually correlates with oxidation rate as well (better/higher COD removal would result in more detoxification/a reduction in toxicity).

Answer: Thank you for this consideration. The reviewer is right. However, during the oxidation reaction there is the possibility of forming more toxic by-products than the initial contaminants. In this case, the COD would be lower, but the toxicity should be higher. So, although that rule could be applied, it is not necessarily true in all the cases. For the present case, the statement provided by the reviewer occurs. This is, for higher COD removals, there is a decrease in the toxicity. Nevertheless, when the PDADMAC is used as coagulant, there is a decrease of COD but an increase in toxicity (although by the addition of a toxic chemical from the coagulant rather than due to the formation of a reaction by-product).

RW#1: Results in Table 3: A difference (change, decrease) in GI from 92% to 89% after adsorption and Fenton’s reagent is not so significant. It indicates that the toxicity does not re-increase.

Answer: Thank you for this consideration. In this case it seems that there is no formation of high toxic by-products although the COD decreases after applying the Fenton reaction. However, in the case of PDADMAC coagulation, the germination index changes from 48% (strong inhibition) to 92% (no inhibition), and in this case, it is possible to see that the Fenton was important to remove toxic contaminants from the (pre-treated) wastewater. The following information was introduced in the manuscript (section 3.3.3, lines 468-472).

“[...]. When the RM pretreated wastewater was considered, the samples did not show toxicity, although a negligible GI decrease from 95% to 89% was observed. This shows that the toxicity does not increase, so it can be assumed that no relevant toxic by-products were produced during the Fenton process application.

RW#1: Tables (1-2-3) could be removal into an Appendix or Supplementary Section. In the main manuscript text, figures should be preferred (for Tables 2-3 ,not Table 1) since these are much easier to follow and interpret (also visually more attractive).

Answer: Thank you for this consideration. All the data was combined into only one table.

Table 1. Characterization of treated wastewater after the different processes.

Parameter

Initial value

Fenton after PDADMAC coagulation

Fenton after RM adsorption

COD (mgO2/L)

1700

49-264

(850)1

10-171

(240)1

BOD5 (mgO2/L)

219

4.4

30.9

TKN (mg/L)

1706

87.6

175.1

Phosphorous (mg/L)

245

0.0

0.3

Total Iron (mg/L)

12

58.5

55.3

L. sativum GI at 48 h

54.9 / Strong inhibition

92 / No inhibition

(47.5 / Strong inhibition)2

88.7 / No inhibition

(95 / No inhibition)2

1 COD values and, 2 GI results obtained only by the application of pre-treatment (PDADMAC coagulation/RM adsorption)

Conclusion:

RW#1: This work demonstrated that industrial residues can be effectively used as materials for industrial wastewater treatment in a continuous mode of operation. It could be shown that IF is an attractive source of Fe catalyst material for Fenton treatment of SW. Moreover, the study revealed that RM is a potential (even superior) adsorbent and alternative to coagulation with PDADMAC. The toxicity of the original and treated SW was also evaluated using L. Sativum as the test plant of the bioassay. Apparently, SW exhibited a “strong inhibition”, but after pretreatment and Fenton treatment it changed/decreased to a “non-inhibitory” effect. Adsorption proved to be more attractive not only in terms of pollutant removal but also in terms of ecotoxicological effect as compared to coagulation.

Answer: Thank you for your important contribution on the improvement of this manuscript. Several changes were made which improved the quality of the reported work.

Reviewer 2 Report

Comments and Suggestions for Authors

In this paper, the authors propose a method to treat wastewater with high COD load from pig farms by reusing of some industrial waste.

The work generals good, the results are systematically written, and the conclusions are consistent with the paper content.

I have just few minor suggestions:

- Lines: 30-32. The paragraph is unclear. For example “China counting approximately 450 million” pigs or tons of meat?

- Several sentences are too lengthy, making the text difficult to follow, especially in the introduction section. It is recommended to restructure the text and conduct a minor language check.

-Line 278 - Why does the amount of organic compounds increase with the partial addition of hydrogen peroxide?

- Line 327 - In Figure 4, what does "(adi)" mean?

Comments on the Quality of English Language

Minor editing of English language required!

Author Response

Reviewer #2 (RW#2):

RW#2: In this paper, the authors propose a method to treat wastewater with high COD load from pig farms by reusing of some industrial waste. The work generals good, the results are systematically written, and the conclusions are consistent with the paper content. I have just few minor suggestions:

- Lines: 30-32. The paragraph is unclear. For example “China counting approximately 450 million” pigs or tons of meat?

Answer: Thank you for this consideration. This was revised and changed to (section 1, line 32): “[…] China counting approximately 450 million pigs […]”.

RW#2: - Several sentences are too lengthy, making the text difficult to follow, especially in the introduction section. It is recommended to restructure the text and conduct a minor language check.

Answer: We acknowledge your opinion. Several changes in the manuscript language were carried.

RW#2: -Line 278 - Why does the amount of organic compounds increase with the partial addition of hydrogen peroxide?

Answer: Thank you for this consideration. We revised the results and verified that no substantial change occurred after the partial addition of H2O2. Therefore, this section was deleted and the following sentence related to these results were added to section 3.1.1. (lines 319-326):

“[…]. Experiments involving additional H2O2 dosing along the reaction time were performed. After the first Fenton reaction ([IF] = 15 g/L, [H2O2] = 50 mg/L, pH = 3, time = 60 min), another dose of H2O2 was added (25, 50 or 500 mg/L) and the reaction continued during further 60 min. The COD removal did not substantially change after the additional dose, which suggests that after the initial 60 min there is still traces of H2O2 and that the reaction is not limited by the H2O2 consumption. In fact, Le et al., [65] was able to increase the degradation in only 10% by adding partial H2O2, which show that the partial H2O2 dosing may not be an efficient way of increasing the organic contaminant’s degradation.”.

RW#2: - Line 327 - In Figure 4, what does "(adi)" mean?

Answer: Thank you for the pertinent question. It should be written the diminutive of “dimensionless”. Changes were made and “adi” was substituted by “(-)”.

Reviewer 3 Report

Comments and Suggestions for Authors

The manuscript entitled “Continuous heterogenous Fenton for swine wastewater treatment: Converting an industry waste into a wastewater treatment material” investigate the combination of treatment processes to remove basic physico-chemical parameters such as COD, BOD, TKN, total phosphorus. The efficiency of experiments is based on COD removal. The advantage of this paper is that real sample, swine wastewater is used for experiments and the second part of experiments is performed in continuous mode using column experiments for Fenton process. The first part of the experiments based on comparison of adsorption with red mud with coagulation with commercial coagulant is performed in batch mode. The advantage is also usage of waste materials in treatment process which improves sustainable development and aims to circular economy. The introduction presents the current state of the art in the area and is well described.

The last paragraph in introduction which presents the importance of the current paper needs to be improved. The importance of this study with all aims – global and specific needs to be added in this last paragraph of introduction.

The material and methods part is well described. The part which explains the preparation of raw waste materials such as iron filings and red mud needs to be explained in detail. Also in analytical part is missing QA/QC.

In addition, authors performed toxicity evaluations and analysis of treated wastewater and compared results with the values prescribed by their national regulations.

The authors mention in part results and discussion which parameters need to be improved to achieve an economic friendly wastewater treatment and reduce operational costs for industrial applications of suggested treatments. However, the part conclusions need to be rewritten to obtain the main advantages and limitations of suggested treatment and possible recommendations for future investigation.

Author Response

Reviewer #3 (RW#3):

(RW#3): The manuscript entitled “Continuous heterogenous Fenton for swine wastewater treatment: Converting an industry waste into a wastewater treatment material” investigate the combination of treatment processes to remove basic physico-chemical parameters such as COD, BOD, TKN, total phosphorus. The efficiency of experiments is based on COD removal. The advantage of this paper is that real sample, swine wastewater is used for experiments and the second part of experiments is performed in continuous mode using column experiments for Fenton process. The first part of the experiments based on comparison of adsorption with red mud with coagulation with commercial coagulant is performed in batch mode. The advantage is also usage of waste materials in treatment process which improves sustainable development and aims to circular economy. The introduction presents the current state of the art in the area and is well described.

Answer: Thank you so much for your words.

(RW#3): The last paragraph in introduction which presents the importance of the current paper needs to be improved. The importance of this study with all aims – global and specific needs to be added in this last paragraph of introduction.

Answer: We acknowledge your opinion. The last paragraph from “introduction” section was reviewed and changed accordingly (lines 177-185):

“In this work, the main objective was to investigate the use of waste materials in the treatment of real swine wastewater, targeting a circular economy approach. For this, adsorption with red mud was evaluated as a potential substitute for PDADMAC coagulation as a pre-treatment. Then, the efficiency of iron fillings waste as the iron source in Fenton’s process was investigated. The Fenton process was effectively applied in continuous operation mode using a very low concentration of H2O2, which should motivate the industrial application of such technology. To the best of our knowledge, no other work reports the treatment of real swine wastewater in continuous operation using RM and IF as raw materials.”.

(RW#3): The material and methods part is well described. The part which explains the preparation of raw waste materials such as iron filings and red mud needs to be explained in detail. Also, in analytical part is missing QA/QC.

Answer: Thank you so much for your note. The pretended information was added to the manuscript (section 2.1), as the standard error deviation of analytical measurements (section 2.2, lines 221 and 222).

Section 2.1: “In this work, two different materials were used for the SW treatment. The methodologies for the preparation and characterization of the IF, RM and SW are presented elsewhere [36, 57]. 

In general, the IF was obtained by disassembling iron wastes from construction industry, while the RM was provided by an aluminum oxide industry located in Greece and sieved (0.105 mm) before being used. The RM presents a red color, a pHzpc (pH zero point of charge) of 13 [57], a surface area of 0.6 m2/g [57], and constituted by several metals (Cu, Zn, Fe, Cr, Ni, Mn, Pb, Mg and Al) with the Fe and Al being the most prevalent. The IF presented a specific surface area of 1.14 m2/g and an average pore diameter of 4.43 nm, and it is practically only composed by iron [58].

The SW was collected from a pig farm located in the center region of Portugal and posteriorly diluted with distilled water (8 vol%) to simulate the wastewater that is resultant from the pig farm washing. The SW presented a chemical oxygen demand (COD) of 1700 mgO2/L, biochemical oxygen demand at day 5 (BOD5) of 219 mgO2/L, total solids (TS), Kjeldahl nitrogen (TKN), phosphorous of 1706 mg/L, 245 mg/L, and 12 mg/L respectively, pH of 7.5, and a zeta potential of -19 mV (pH = 7) and -10 mV (pH = 3) [36].”.

Section 2.2: “[…].  The experiments were made in duplicate and the standard deviations are presented in all the Figures as error bars.”.

(RW#3): In addition, authors performed toxicity evaluations and analysis of treated wastewater and compared results with the values prescribed by their national regulations.

Answer: Thank you so much for your comment. We used the Portuguese decrees of law as metrics and guidelines to evaluate the general wastewater treatment efficiency. This is important, because it can help to understand if the reclaimed water is prepared to be discharged into the water resources or if more treatments are required. So, this analysis can indirectly evaluate the treatment efficiency. For this case, it was concluded that other treatments are necessary. There is the need of removing excessive (dissolved) iron, and the biological treatments should not also be discarded from the global treatment train. To going further from national regulation as considering that in the future such legislation will be stricter and consider the impact of the treated wastewater in health and ecosystems, the samples toxicity along the treatment train was also evaluated.

(RW#3): The authors mention in part results and discussion which parameters need to be improved to achieve an economic friendly wastewater treatment and reduce operational costs for industrial applications of suggested treatments. However, the part conclusions need to be rewritten to obtain the main advantages and limitations of suggested treatment and possible recommendations for future investigation.

Answer: We acknowledge your opinion. With the methodology followed in this work, it is possible to withdraw the following advantages: i) contribution to circular economy by using industrial wastes, ii) reduce the operational costs due to the use of residues, iii) have an efficient treatment using Fenton reaction in continuous operation and iv) motivate the industrial application of this technology due to high efficiency in continuous and associated low-cost.

Therefore, the conclusion was revised, and the following sentences were added (conclusion, lines 497-507):

“[…]. Therefore, the advantages of this work are the efficient use of wastes as raw materials (in which the RM adsorption can be effectively used as PDADMAC coagulation alternative), allowing to decrease the treatment costs, contributing to the circular economy, and having high efficiency for the treatment of real wastewater. As drawbacks, the iron filings will need to be replaced after some operation time and the fact that other treatments are required to remove iron and ammonium/ammonia. The application of biological treatments can be a suitable solution for this purpose.

For future work, different types of real wastewater should be tested, and scale-up studies should be carried. The use of photo-Fenton instead of Fenton, namely using solar radiation could also be an interesting approach and should not be discarded. Also, toxicity studies involving other trophic level species should be carried out.”.

Reviewer 4 Report

Comments and Suggestions for Authors

I revised the work in which the authors applied a continuous heterogenous Fenton in order to treat swine wastewater. The manuscript is well written, but amendments should be implemented before a possible publication. My comments are the following:

·       Lines 123-124: please, try to describe better this aspect highlighting the aims of your work.

·       Section 2.1. The initial characteristics of WW must be reported in the text.

·       Figure 2. A suggestion. Why don’t plot this data as a scatter plot also on x-axis? In this way you can highlight potential correlation between COD removal and H2O2 addition.

·       Figures 1,2, and 3. Please, in the caption define if the bars are the confidence intervals or the standard deviations and the number of data.

·       Table 3: “54.9 / Strong inhibition” is referred to what column? “PDADMAC coagulation experiment” or “RM adsorption experiment” or both? In this last case, please repeat in both columns. Now it is not clear.

·       Please, before the conclusions include a section reporting the limitations of your work (e.g., the use of synthetic WW).

·       You use a “synthetic” WW. Please, discuss how the presence of suspended organic matter in real swine WW could influence the results of application of this process.

Author Response

Reviewer #4 (RW#4):

(RW#4): I revised the work in which the authors applied a continuous heterogenous Fenton in order to treat swine wastewater. The manuscript is well written, but amendments should be implemented before a possible publication. My comments are the following:

(RW#4): Lines 123-124: please, try to describe better this aspect highlighting the aims of your work.

Answer: We acknowledge your opinion. Changes were made, and this last paragraph was revised (introduction, lines 177-185) to:

“In this work, the main objective was to investigate the use of waste materials in the treatment of real swine wastewater, targeting a circular economy approach. For this, adsorption with red mud was evaluated as a potential substitute for PDADMAC coagulation as a pre-treatment. Then, the efficiency of iron fillings waste as the iron source in Fenton’s process was investigated. The Fenton process was effectively applied in continuous operation mode using a very low concentration of H2O2, which should motivate the industrial application of such technology. To the best of our knowledge, no other work reports the treatment of real swine wastewater in continuous operation using RM and IF as raw materials.”.

(RW#4): Section 2.1. The initial characteristics of WW must be reported in the text.

Answer: Thank you for your commentary. All this section was revised, and the final text is present:

Section 2.1: “In this work, two different materials were used for the SW treatment. The methodologies for the preparation and characterization of the IF, RM and SW are presented elsewhere [36, 57]. 

In general, the IF was obtained by disassembling iron wastes from construction industry, while the RM was provided by an aluminum oxide industry located in Greece and sieved (0.105 mm) before being used. The RM presents a red color, a pHzpc (pH zero point of charge) of 13 [57], a surface area of 0.6 m2/g [57], and constituted by several metals (Cu, Zn, Fe, Cr, Ni, Mn, Pb, Mg and Al) with the Fe and Al being the most prevalent. The IF presented a specific surface area of 1.14 m2/g and an average pore diameter of 4.43 nm, and it is practically only composed by iron [58].

The SW was collected from a pig farm located in the center region of Portugal and posteriorly diluted with distilled water (8 vol%) to simulate the wastewater that is resultant from the pig farm washing. The SW presented a chemical oxygen demand (COD) of 1700 mgO2/L, biochemical oxygen demand at day 5 (BOD5) of 219 mgO2/L, total solids (TS), Kjeldahl nitrogen (TKN), phosphorous of 1706 mg/L, 245 mg/L, and 12 mg/L respectively, pH of 7.5, and a zeta potential of -19 mV (pH = 7) and -10 mV (pH = 3) [36].”.

(RW#4): Figure 2. A suggestion. Why don’t plot this data as a scatter plot also on x-axis? In this way you can highlight potential correlation between COD removal and H2O2 addition.

Answer: We acknowledge your opinion. We revised the results and verified that no substantial change occurred after the partial addition of H2O2. Therefore, this section was deleted and the following sentence related to these results were added to section 3.1.1. (lines 319-326):

“[…]. Experiments involving additional H2O2 dosing along the reaction time were performed. After the first Fenton reaction ([IF] = 15 g/L, [H2O2] = 50 mg/L, pH = 3, time = 60 min), another dose of H2O2 was added (25, 50 or 500 mg/L) and the reaction continued during further 60 min. The COD removal did not substantially change after the additional dose, which suggests that after the initial 60 min there is still traces of H2O2 and that the reaction is not limited by the H2O2 consumption. In fact, Le et al., [65] was able to increase the degradation in only 10% by adding partial H2O2, which show that the partial H2O2 dosing may not be an efficient way of increasing the organic contaminant’s degradation.”.

(RW#4): Figures 1, 2, and 3. Please, in the caption define if the bars are the confidence intervals or the standard deviations and the number of data.

Answer: Thank you for your note. The bars are the standard deviations of the presented data. To don’t turn the Figures caption very extensive, a note for this was added to the section 2.2 (lines 221 and 222).

“[…]. The experiments were made in duplicate and the standard deviations are presented in all the Figures as error bars.”

(RW#4): Table 3: “54.9 / Strong inhibition” is referred to what column? “PDADMAC coagulation experiment” or “RM adsorption experiment” or both? In this last case, please repeat in both columns. Now it is not clear.

Answer: Thank you for your consideration. The value is referred to both columns. However, the Tables were revised accordingly to all the reviewer’s suggestions.

(RW#4): Please, before the conclusions include a section reporting the limitations of your work (e.g., the use of synthetic WW).

Answer: We acknowledge your opinion. Sorry for the misunderstanding, but in our work, it was only used real swine wastewater.

However, the use of synthetic wastewater would need a new Fe load and H2O2 concentration optimization. Probably better results would be obtained if a synthetic wastewater was applied due to the non-existence of inorganic matter that may act as •OH radical’s scavengers (as SO4-, HCO3- or CO32-) that are present in real wastewaters.

(RW#4): You use a “synthetic” WW. Please, discuss how the presence of suspended organic matter in real swine WW could influence the results of application of this process.

Answer: We acknowledge your opinion, and we are sorry for the misunderstanding. In this work, all the experiments used real wastewater and no synthetic wastewater was considered. As already referred, the organic matter should have •OH radical’s scavengers which should decrease the process performance. The real effluent indeed contained suspended organic matter that was removed using two different approaches: coagulation with a commercial coagulant and adsorption using the red mud waste.

Round 2

Reviewer 1 Report

Comments and Suggestions for Authors

The authors have addressed most of the answers and comments in detail. The manuscript has been improved appreciably. A minor comment: Please check for some typos and be sure that the Conclusion does not repeat the Abstract. Thank you.

Comments on the Quality of English Language

Has already mentioned in the above parts of the review.

Author Response

Reviewer #1 (RW#1):

RW#1: The authors have addressed most of the answers and comments in detail. The manuscript has been improved appreciably. A minor comment: Please check for some typos and be sure that the Conclusion does not repeat the Abstract. Thank you.

 Answer: We acknowledge your commentary. Changes were made accordingly. Thank you for your advises and comments that allowed to improve the quality of this manuscript.

Reviewer 4 Report

Comments and Suggestions for Authors

Thank you for your implementations.

Author Response

Reviewer #4 (RW#4):

RW#4: Thank you for your implementations. 

Answer: We appreciate and are grateful for your important advice and comments, which helped to improve the quality of this manuscript. Thank you.
